# FcγRIIb-SHIP2 axis links Aβ to tau pathology by disrupting phosphoinositide metabolism in Alzheimer's disease model

Tae-In Kam[1,2,3†], Hyejin Park[1,2,3†], Youngdae Gwon[1†], Sungmin Song[1], Seo-Hyun Kim[1], Seo Won Moon[1], Dong-Gyu Jo[4], Yong-Keun Jung[1]*

[1]Global Research Laboratory, School of Biological Sciences, Seoul National University, Seoul, Korea; [2]Neuroregeneration and Stem Cell Programs, Institute for Cell Engineering, Baltimore, United States; [3]Department of Neurology, Johns Hopkins University School of Medicine, Baltimore, United States; [4]School of Pharmacy, Sungkyunkwan University, Suwon, Korea

**Abstract** Amyloid-β (Aβ)-containing extracellular plaques and hyperphosphorylated tau-loaded intracellular neurofibrillary tangles are neuropathological hallmarks of Alzheimer's disease (AD). Although Aβ exerts neuropathogenic activity through tau, the mechanistic link between Aβ and tau pathology remains unknown. Here, we showed that the FcγRIIb-SHIP2 axis is critical in $Aβ_{1-42}$-induced tau pathology. *Fcgr2b* knockout or antagonistic FcγRIIb antibody inhibited $Aβ_{1-42}$-induced tau hyperphosphorylation and rescued memory impairments in AD mouse models. FcγRIIb phosphorylation at Tyr273 was found in AD brains, in neuronal cells exposed to $Aβ_{1-42}$, and recruited SHIP2 to form a protein complex. Consequently, treatment with $Aβ_{1-42}$ increased PtdIns$(3,4)P_2$ levels from PtdIns$(3,4,5)P_3$ to mediate tau hyperphosphorylation. Further, we found that targeting SHIP2 expression by lentiviral siRNA in 3xTg-AD mice or pharmacological inhibition of SHIP2 potently rescued tau hyperphosphorylation and memory impairments. Thus, we concluded that the FcγRIIb-SHIP2 axis links Aβ neurotoxicity to tau pathology by dysregulating PtdIns$(3,4)P_2$ metabolism, providing insight into therapeutic potential against AD.

*For correspondence: ykjung@snu.ac.kr

†These authors contributed equally to this work

Competing interests: The authors declare that no competing interests exist.

## Introduction

Alzheimer's disease (AD) is characterized by the progressive loss of memory and the neuronal degeneration (*Mattson, 2004*). The pathological hallmarks of AD are the presence of senile plaques consisting of Aβ peptide and neurofibrillary tangles (NFT) formed by abnormally hyperphosphorylated tau (*Walsh and Selkoe, 2004*). Aβ species are generated from amyloid precursor protein (APP) by β- and γ-secretases, and accumulate extracellularly (*Haass and Selkoe, 2007*). In general, Aβ contributes to AD pathology by exhibiting toxicity in susceptible neurons, facilitating tau hyperphosphorylation, disrupting proteasome activity, and triggering synaptic dysfunction (*LaFerla et al., 2007*; *Kam et al., 2014*). Tau is a microtubule-binding protein but dissociates from the microtubules and accumulates in neurons as it becomes highly phosphorylated at multiple sites, resulting in the impairment of microtubule assembly and function (*Ballatore et al., 2007*).

Accumulating evidence strongly indicate that these two hallmarks are strongly interrelated in AD. In the amyloid cascade hypothesis, tau is believed to be one of the major downstream targets of Aβ to produce neurotoxicity (*Hardy and Selkoe, 2002*). Aβ accelerates neurodegeneration in neuronal cells but not in tau-deficient neurons (*Rapoport et al., 2002*). Moreover, tau depletion in mutant *APP* transgenic mice prevented Aβ pathologies, including learning and memory impairment (*Roberson et al., 2007*). The role of Aβ in tau pathology was also shown in 3xTg-AD mice

**eLife digest** In Alzheimer's disease, damage to neurons in the brain gradually causes memory loss and difficulties with thinking. The main hallmarks of this damage are seen in the accumulation of proteins in and around neurons. First, a protein called amyloid beta forms aggregates outside the cell. This appears to lead to the build up of an abnormal form of a protein called tau inside the cell. These abnormal tau proteins have excessive numbers of phosphate groups attached to them, and so are known as "hyperphosphorylated". The molecular mechanism underlying amyloid beta's role in the hyperphosphorylation of tau proteins was not known.

Amyloid beta binds to many different receptor proteins – including one called Fc gamma receptor IIb – on the surface of neurons. Kam, Park et al. investigated whether interactions between amyloid beta and Fc gamma receptor IIb might regulate the phosphorylation of tau within neurons. Adding amyloid beta to mouse neurons caused tau proteins to become hyperphosphorylated. However, removing Fc gamma receptor IIb from the neurons, or stopping it from binding to amyloid beta, abolished this effect.

When amyloid beta was bound to Fc gamma receptor IIb, the receptor became phosphorylated. This in turn triggered a series of further phosphorylation events, culminating in an increase in the level of a molecule that relays signals from cell receptors – called SHIP2 – inside the neurons. This molecule increases tau phosphorylation when added to neurons. Reducing the activity or amount of SHIP2 in mice that present the symptoms of Alzheimer's disease reduced the hyperphosphorylation of the tau protein in their neurons and restored their memory to normal levels.

Kam, Park et al. also looked at samples taken from the brains of human Alzheimer's disease patients. Unlike samples taken from people without Alzheimer's disease, neurons in these samples contain both phosphorylated Fc gamma receptor IIb and hyperphosphorylated tau proteins.

By uncovering the molecules that link amyloid beta with tau hyperphosphorylation, Kam, Park et al.'s results suggest new targets for therapies to treat the symptoms of Alzheimer's disease. More research is now needed to investigate whether this could lead to the design of new drugs.

expressing APP, presenilin, and tau transgenes in which Aβ immunization reduced not only Aβ accumulation but also tau pathology (*Oddo et al., 2004*). In addition, higher levels of NFT have been observed in APPswe/P301L transgenic mice (*Lewis et al., 2001*) and in 3xTg-AD mice (*Oddo et al., 2003*). More importantly, tau hyperphosphorylation is frequently found in AD brains (*Grundke-Iqbal et al., 1989*). Apparently, tau kinases, such as glycogen synthase kinase-3β (GSK-3β), are activated by Aβ for tau phosphorylation in vitro and in vivo (*Hoshi et al., 2003*; *Ma et al., 2006*; *Terwel et al., 2008*; *Park et al., 2012*). All of these findings indicate the presence of a pathologic signal pathway starting with extracellular Aβ and ending in the phosphorylation of intracellular tau. However, the mechanism connecting the two pathologic hallmarks of AD remains unknown.

Phosphoinositides, the phosphorylated derivatives of phosphatidylinositol (PtdIns), such as PtdIns$(3,4,5)P_3$, PtdIns$(4,5)P_2$, and PtdIns$(3,4)P_2$, are known to play a major role in signal transduction upon cellular stimulation (*Di Paolo and De Camilli, 2006*). Among them, the biological roles of PtdIns$(3,4,5)P_3$ and PtdIns$(4,5)P_2$ have been relatively well characterized in cell survival, proliferation, and synaptic function via their binding proteins (*Bunney and Katan, 2010*; *Khuong et al., 2013*), but the function of PtdIns$(3,4)P_2$ is largely unknown. Unlike PtdIns$(4,5)P_2$, PtdIns$(3,4)P_2$ and PtdIns$(3,4,5)P_3$ are formed when cells respond to signals (*Zhang and Majerus, 1998*; *Lemmon, 2008*). SH2 domain-containing phosphatidylinositol 5'-phosphatase (SHIP) removes 5' phosphate from PtdIns$(3,4,5)P_3$ to produce PtdIns$(3,4)P_2$ (*Damen et al., 1996*). Increasing evidence has revealed that phosphoinositide metabolism is dysregulated in AD; specifically, the level of PtdIns$(4,5)P_2$ is decreased in human and mouse AD brains, and in the primary cortical neurons exposed to oligomeric Aβ (*Stokes and Hawthorne, 1987*; *Jope et al., 1994*; *Berman et al., 2008*), and recovery of PtdIns$(4,5)P_2$ deficiency prevents AD-related cognitive deficits in mouse models (*McIntire et al., 2012*; *Zhu et al., 2015*). However, how phosphoinositide metabolism, including levels of PtdIns$(3,4)P_2$, is regulated by Aβ during AD pathogenesis and the consequences of its dysregulation in AD needs to be resolved.

Until now, Aβ was reported to bind to many receptors, including alpha7 nicotinic acetylcholine receptors (α7 nAChR), NMDA receptor, receptors for advanced glycation end- products (RAGE), Aβ-binding alcohol dehydrogenase (ABAD), the Ephrin-type B2 receptor (EphB2), cellular prion protein (PrPc), and paired immunoglobulin-like receptor B (PirB) (*Yan et al., 1996*; *Wang et al., 2000*; *Lustbader, 2004*; *Snyder et al., 2005*; *Laurén et al., 2009*; *Cissé et al., 2011a*; *Kim et al., 2013*). Although these receptors were shown to be responsible for Aβ neurotoxicity, especially memory impairment in AD mice, their role as neuronal receptors in Aβ-induced tau pathologies was limitedly shown in α7 nAChR and NMDA receptor (reviewed in *Stancu et al., 2014*). Of particular note, while α7 nAChR was reported to mediate Aβ-induced tau phosphorylation, the finding was based on in vitro and ex vivo system (*Wang et al., 2003*). Furthermore, evidence showing a correlation of the proposed molecular mechanism with pathologic evidence was not much provided. In particular, the CAMKK2-AMPK at down-stream of NMDA receptor was recently proposed to mediate the synapto-toxic effects of Aβ oligomers through tau phosphorylation and this event is very likely caused by NMDA receptor-induced increase of intracellular calcium, not by direct interaction of NMDA receptor with Aβ (*Mairet-Coello et al., 2013*). Therefore, a neuronal receptor that is important in Aβ-induced tau pathology needs to be elucidated.

Recently, we showed that Fc gamma receptor IIb (FcγRIIb) is also expressed in neurons and directly interacts with $A\beta_{1-42}$ to mediate Aβ neurotoxicity, synaptic dysfunction, and memory impairment in AD pathogenesis (*Nimmerjahn and Ravetch, 2008*; *Kam et al., 2013*). Here, we show that FcγRIIb is phosphorylated at tyrosine 273 by $A\beta_{1-42}$ in neurons and in AD brains, and that this phosphorylation recruits SH2 domain-containing phosphatidylinositol 5'-phosphatase 2 (SHIP2, INPPL1) to increase $PtdIns(3,4)P_2$ levels for tau hyperphosphorylation. Further, *Fcgr2b* or *Inppl1* deficiency in 3xTg-AD mice or pharmacological inhibition of either protein abrogates all of these observations, highlighting the importance of the FcγRIIb-SHIP2 axis in the Aβ-induced tau pathology.

## Results

### FcγRIIb is essential for tau hyperphosphorylation and memory deficit in 3xTg-AD mice

Given that FcγRIIb was previously identified as a receptor for $A\beta_{1-42}$ and a mediator of memory impairment in hAPP-J20 mice expressing only familial mutant APP (*Kam et al., 2013*), and that tau as well as Aβ is essential for memory impairment in AD mouse (*Ramsden et al., 2005*), we hypothesized that FcγRIIb is responsible for tau hyperphosphorylation in response to $A\beta_{1-42}$. We first assessed and confirmed that both FcγRIIb mRNAs and proteins were significantly expressed in neurons as well as non-neuronal cells in mouse brains (*Figure 1—figure supplement 1*). Interestingly, incubation of primary cortical neurons with synthetic $A\beta_{1-42}$ oligomers increased tau phosphorylation at several pathological epitopes, including Ser396/Ser404 (PHF1), Thr231/Ser235 (AT180), and Ser202 (CP13), but these phosphorylations were abrogated in *Fcgr2b* knockout (KO) neurons (*Figure 1A,B*). Immunocytochemical analysis also showed that immunoreactivity against the phosphorylated tau was increased by treatment with $A\beta_{1-42}$ in the neuron-specific enolase (NSE)-positive cortical neurons, but not in the *Fcgr2b* KO neurons (*Figure 1C*).

Instead of using a high dose of synthetic $A\beta_{1-42}$, we further utilized a low-dose of cell-derived, naturally secreted Aβ oligomers (*Walsh et al., 2002*). The presence of the secreted, soluble Aβ oligomers in the conditioned medium (CM) of 7PA2 cells was confirmed by western blot analysis, and its production was inhibited by treatment with a γ-secretase inhibitor, compound E (*Figure 1—figure supplement 2A*). We found that tau phosphorylation was increased in the primary hippocampal neurons that were cocultured with 7PA2 cells (*Figure 1D*), but not in the neurons cocultured with compound E-treated cells (*Figure 1—figure supplement 2B*). Further, we found that there was no hyperphosphorylation of tau in the *Fcgr2b* KO neurons that were cocultured with 7PA2 cells (*Figure 1D*). These results suggest that tau hyperphosphorylation in the cultured neurons was induced by physiologically relevant levels of Aβ oligomers through FcγRIIb.

The 3xTg-AD mice develop age-dependent and progressive Aβ and tau pathologies, including memory impairment (*Oddo et al., 2003*). Thus, in order to uncover the role of FcγRIIb in tau pathology, we crossed 3xTg-AD mice with *Fcgr2b* KO mice to generate double transgenic mice (3xTg-AD/*Fcgr2b* KO). Each group of mice was tested for spatial working memory in a Y-maze task. Although

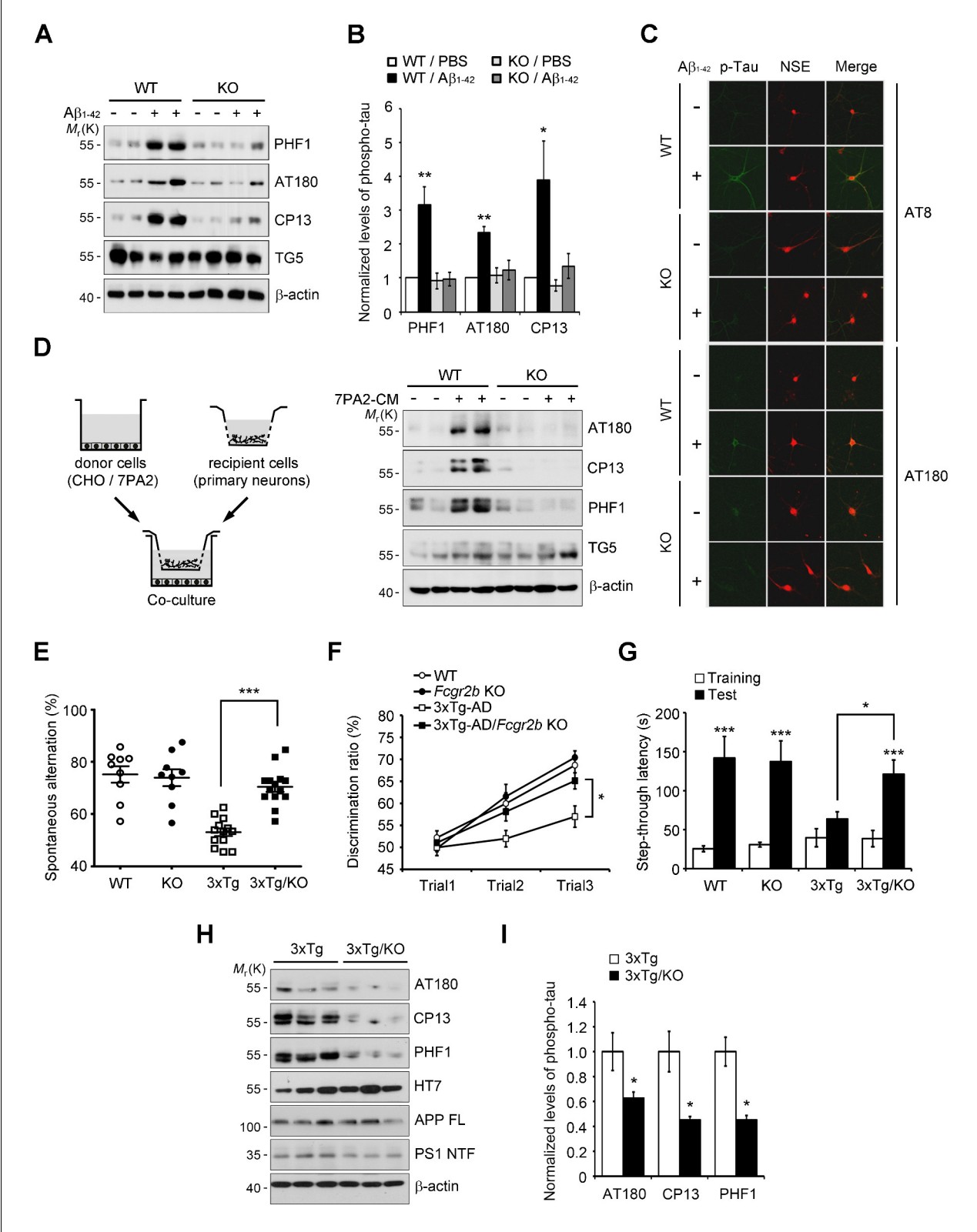

**Figure 1.** *Fcgr2b* deficiency prevents tau hyperphosphorylation and memory deficits in 3xTg mice. (**A**, **B**) *Fcgr2b* KO neurons are resistant to Aβ-induced tau phosphorylation. Mouse primary cortical neurons from wild-type (WT) or *Fcgr2b* KO embryos (DIV 8) were incubated with oligomeric forms of 1 μM synthetic Aβ$_{1-42}$ for 24 hr and cell extracts were subjected to western blotting (**A**). The levels of phosphorylated tau were quantified by densitometry and normalized by total tau (TG5). Values are means ± s.e.m.; *p<0.05, **p<0.005, one-way ANOVA (n = 4) (**B**). (**C**) Immunocytochemical

*Figure 1 continued on next page*

*Figure 1 continued*

analysis of Aβ-induced tau phosphorylation in WT and *Fcgr2b* KO neurons. (D) *Fcgr2b* deficiency prevents cell-derived Aβ-induced hyperphosphorylation of tau. Mouse primary hippocampal neurons from WT or *Fcgr2b* KO embryos were cocultured with CHO or 7PA2 cells for 24 hr (*left*) and tau phosphorylation was analyzed by western blotting (*right*). (E–F) Rescue of memory impairment in 3xTg-AD/*Fcgr2b* KO mice. Y-maze (E), novel object recognition (F), and passive avoidance (G) tests were performed in 8–9 month-old WT, *Fcgr2b* KO, 3xTg-AD, and 3xTg-AD/*Fcgr2b* KO mice (n = 9–14 mice per group; WT, 5 males and 4 females; KO, 5 males and 4 females; 3xTg-AD, 8 males and 5 females; 3x-Tg-AD/KO, 7 males and 7 females). Data are means ± s.e.m.; *p<0.05, ***p<0.001, unpaired *t*-test. (H, I) Reduced hyperphosphorylation of tau in 3xTg-AD/*Fcgr2b* KO mice. The hippocampal lysates of 9 month-old mice were subjected to western blotting (H). The levels of phosphorylated tau were quantified as in (A). Values are means ± s.e.m.; *p<0.05, unpaired *t*-test (n = 3) (I).

The following figure supplements are available for figure 1:

**Figure supplement 1.** Neuronal expression of FcγRIIb in the mouse brain.

**Figure supplement 2.** FcγRIIb is required for cell-derived Aβ oligomer-induced tau phosphorylation in the primary neurons.

**Figure supplement 3.** FcγRIIb is also required for Aβ-induced tau phosphorylation in the hippocampus of aged 3xTg-AD and hAPP (J20) mouse lines.

the total number of arm entries was not significantly different between groups, *Fcgr2b* deficiency improved the spatial working memory of 3xTg-AD mice (*Figure 1E*). In the novel object recognition test, only 3xTg-AD/*Fcgr2b* KO mice, but not 3xTg-AD mice, discriminated between novel and familiar objects in the second and third trials (*Figure 1F*), indicating that the recognition memory deficit of 3xTg-AD mice was prevented by *Fcgr2b* deficiency. Further, 3xTg-AD/*Fcgr2b* KO mice performed well in the passive avoidance task, whereas 3xTg-AD mice showed a deficit in passive avoidance memory (*Figure 1G*). These data indicate that FcγRIIb is required for the learning and memory impairments in 3xTg-AD mice.

Because FcγRIIb itself did not affect Aβ levels in these mice (*Figure 1—figure supplement 3A*) and the hippocampus of 3xTg-AD mice at nine months of age was plaque-free (*Hirata-Fukae et al., 2008*), we next examined the role of FcγRIIb in tau pathology in the mouse brains. As reported (*Hirata-Fukae et al., 2008*), the pathologic hyperphosphorylation of tau detected by CP13, PHF1, and AT180 antibodies was found to be increased in the 3xTg-AD mice showing memory impairment (*Figure 1H,I*). In contrast, genetic deletion of *Fcgr2b* in 3xTg-AD mice abolished the hyperphosphorylation of tau. In the brains of 20 month-old 3xTg-AD mice harboring Aβ plaques, tau phosphorylation was markedly elevated as compared to 6 month-old mice (*Figure 1—figure supplement 3B,C*). FcγRIIb deficiency also prevented the hyperphosphorylation of tau in 15 month-old 3xTg-AD mice (*Figure 1—figure supplement 3D,E*), indicating that tau phosphorylation mediated by FcγRIIb occurs at the onset of the disease and lasts to the late stage. Similarly, this inhibitory effect of *Fcgr2b* deficiency on tau phosphorylation was observed in another AD model, hAPP-J20 mice (*Figure 1—figure supplement 3F,G*). These results indicate that FcγRIIb is crucial for tau hyperphosphorylation in AD model mice showing memory impairment.

## Antagonistic FcγRIIb antibody inhibits tau phosphorylation and memory impairment

Because Aβ transduces toxic signals into the neurons via direct interaction with FcγRIIb (*Kam et al., 2013*), we examined whether this interaction is necessary for tau hyperphosphorylation. Compared to Aβ only-treated neurons, addition of the purified hFcγRIIb ectodomain (hFcγRIIb-ED) to culture medium blocked Aβ-induced tau hyperphosphorylation in primary cortical neurons (*Figure 2A,B*). With the notion that both Aβ$_{1-42}$ oligomers and immunoglobulin complexes share the same binding site on FcγRIIb (*Kam et al., 2013*), we found that the incubation of cultured cells with anti-FcγRIIb antibody (2.4G2) blocked the interaction between Aβ$_{1-42}$ and HA-tagged FcγRIIb (*Figure 2—figure supplement 1A,B*). This antibody selectively recognized FcγRIIb but did not crossreact with Aβ$_{1-40}$ or Aβ$_{1-42}$ (*Figure 2—figure supplement 1C*). Consistently, treatment with 2.4G2 antibody drastically decreased Aβ-induced tau hyperphosphorylation in mouse primary cortical neurons in a dose-dependent manner (*Figure 2C,D*). These observations suggest that the interaction between Aβ$_{1-42}$ and FcγRIIb is an initial step for tau phosphorylation in the cultured neurons.

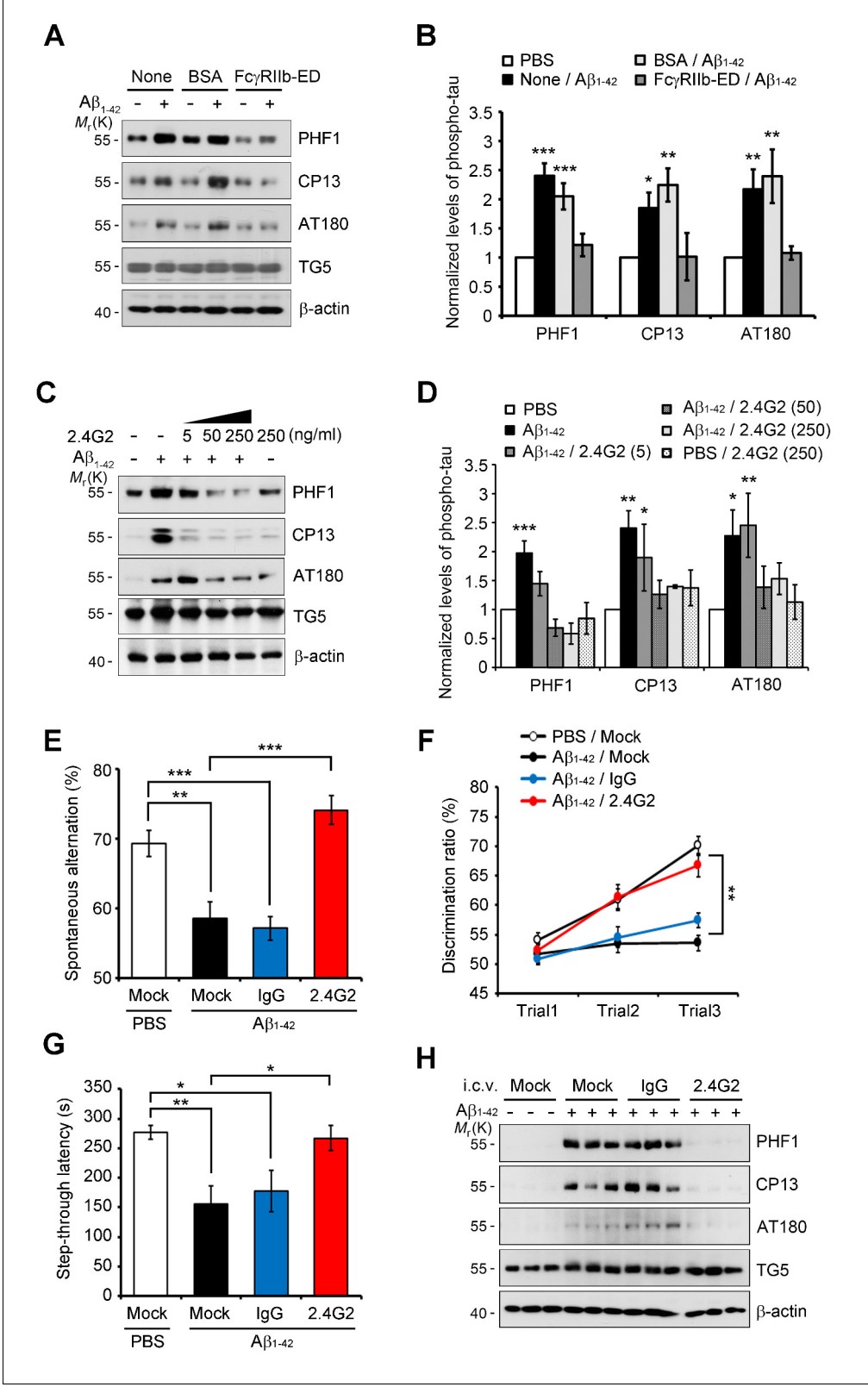

**Figure 2.** Inhibition of Aβ₁₋₄₂-FcγRIIb interaction blocks Aβ-induced tau phosphorylation and memory impairment. (**A, B**) Inhibition of Aβ₁₋₄₂-induced tau phosphorylation by the addition of purified hFcγRIIb-ED protein. Mouse primary cortical neurons were incubated for 24 hr with 1 μM synthetic Aβ₁₋₄₂ w/wo 50 μg/ml purified hFcγRIIb-ED and cell extracts were subjected to western blotting (**A**). The levels of phosphorylated tau (PHF1, CP13, and

*Figure 2 continued on next page*

*Figure 2 continued*

AT180) were normalized by total tau (TG5). Values are means ± s.e.m.; *p<0.05, **p<0.005, ***p<0.0005, one-way ANOVA (n = 3) (B). (C, D) Prevention of Aβ$_{1-42}$-induced tau phosphorylation by anti-FcγRIIb antibody (2.4G2). Mouse primary cortical neurons were pre-incubated with 2.4G2 antibody for 2 hr and treated w/wo 1 μM Aβ$_{1-42}$ oligomers for 24 hr. Cell extracts were analyzed with western blotting using indicated antibodies (C). Levels of phosphorylated tau were quantified by densitometric measurement. Values are means ± s.e.m.; *p<0.05, **p<0.005, ***p<0.0005, one-way ANOVA (n = 3) (D). (E–G) Suppression of Aβ$_{1-42}$-induced cognitive deficits by coinjection of anti-FcγRIIb antibody. WT mice (8 weeks old) were i.c.v.-injected with PBS or Aβ$_{1-42}$ (410 pmol) together w/wo either 2 μg IgG or 2.4G2 antibody. The mice (n = 10 for each group) were analyzed by Y-maze (E; **p<0.005, ***p<0.0005, unpaired *t*-test), novel object recognition (F; **p<0.005, one-way ANOVA), and passive avoidance (G; *p<0.02, **p<0.005, unpaired *t*-test) tests as described in the methods. Bars represent means ± s. e.m. (H) Inhibition of i.c.v. Aβ$_{1-42}$-induced tau phosphorylation by anti-FcγRIIb antibody. Brain extracts of the Aβ- and/or antibody-injected mice were subjected to western blotting.

The following figure supplement is available for figure 2:

**Figure supplement 1.** FcγRIIb-antagonizing antibody, 2.4G2, inhibits the interaction of Aβ$_{1-42}$ with FcγRIIb.

Next, we addressed the effects of 2.4G2 antibody on Aβ-induced acute memory impairment in mice. When Aβ$_{1-42}$ was directly injected into the intracerebroventricular (i.c.v.) region of wild-type (WT) mice (*Kam et al., 2013*), the mice showed impaired behaviors (*Figure 2E–G*). Interestingly, coinjection of Aβ$_{1-42}$ and 2.4G2 antibodies significantly rescued the deficits of spontaneous alternation behavior (*Figure 2E*), object recognition memory (*Figure 2F*), and passive avoidance memory (*Figure 2G*). On the other hand, injection with normal immunoglobulin G (IgG) did not exhibit such inhibitory effects on memory impairment. There was no difference in total movements between these groups of mice, as reflected by total arm entry in the Y-maze test (*Figure 2—figure supplement 1D*). From western blot analysis, we found that tau phosphorylation was highly increased in the hippocampus of Aβ$_{1-42}$-injected mice (*Figure 2H*). In contrast, Aβ-induced tau phosphorylation was abolished in the hippocampus by coinjection with 2.4G2 antibody, but not with IgG (*Figure 2H*). Thus, we found that the interaction between Aβ$_{1-42}$ and FcγRIIb is essential for tau hyperphosphorylation and memory impairment in mice.

## FcγRIIb ITIM phosphorylation found in AD brains is essential for Aβ neurotoxicity and tau phosphorylation

As reported (*Plattner et al., 2006*), we found that tau kinases, such as GSK3β and Cdk5, were activated for tau phosphorylation in cultured neurons by treatment with Aβ$_{1-42}$ (*Figure 3—figure supplement 1A,B*). Notably, *Fcgr2b* deficiency in primary cortical neurons abrogated the activation of GSK3β and tau phosphorylation triggered by Aβ$_{1-42}$, but not that of Cdk5. Conversely, ectopic expression of FcγRIIb induced tau hyperphosphorylation, detected by PHF1, CP13, and AT180 antibodies, in both SH-SY5Y cells and primary cortical neurons (*Figure 3—figure supplement 1C–F*). Consistently, treatment with SB-415286, a GSK3β inhibitor, markedly prevented FcγRIIb-induced tau hyperphosphorylation at those epitopes (*Figure 3—figure supplement 1C,E*), whereas roscovitine, a Cdk5 inhibitor, did not affect tau phosphorylation (*Figure 3—figure supplement 1D,F*). These observations suggest that FcγRIIb transduces Aβ signal into the neuronal cells to activate GSK3β for tau hyperphosphorylation.

We next investigated the mechanism for FcγRIIb transduction of Aβ signal into the neurons for tau phosphorylation, and found that treatment with Aβ$_{1-42}$ induced phosphorylation at Tyr273 within an immunoreceptor tyrosine-based inhibitory motif (ITIM) in the cytosolic region of FcγRIIb in SH-SY5Y cells (*Figure 3A*). This phosphorylation of FcγRIIb was also detected in Aβ-treated primary cortical neurons, but not in *Fcgr2b* KO neurons (*Figure 3B*). Further, addition of purified hFcγRIIb-ED protein prevented Aβ-induced phosphorylation of FcγRIIb-Tyr273 in SH-SY5Y cells (*Figure 3C*), indicating that FcγRIIb is phosphorylated on Tyr273 after its interaction with Aβ$_{1-42}$.

By using FcγRIIb mutants, we further characterized the role of FcγRIIb phosphorylation in neuronal cells. We generated several FcγRIIb mutants lacking the ITIM (ΔITIM) or cytoplasmic region (Δcyto), or substituting the tyrosine residue with phenylalanine (Y273F), and examined their effects on Aβ$_{1-42}$

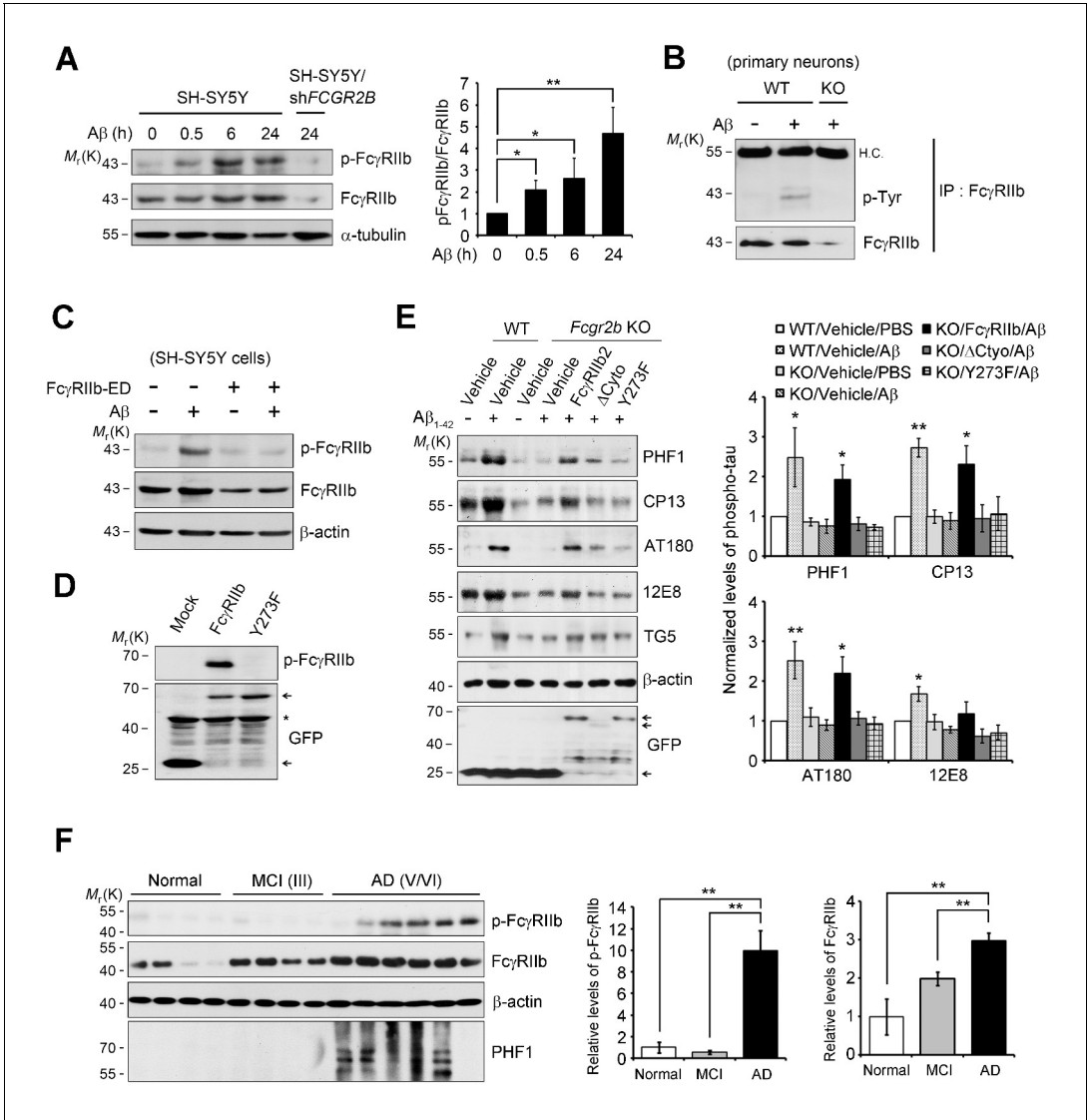

**Figure 3.** FcγRIIb Tyr273 phosphorylation is found in AD brains and mediates Aβ-induced tau phosphorylation. (**A**) Human FcγRIIb is phosphorylated at Tyr273 by Aβ$_{1-42}$. Wild-type and *FCGR2B* knockdown SH-SY5Y cells (SH-SY5Y/sh*FCGR2B*) were incubated with 1 μM Aβ$_{1-42}$ oligomers and cell extracts were subjected to western blotting using phospho-FcγRIIb and total FcγRIIb antibodies (*left*). Levels of the phosphorylated FcγRIIb were quantified by densitometric measurement. Values are means ± s.d.; *p<0.05, **p<0.005, two-tailed *t*-test (n = 3) (*right*). (**B**) FcγRIIb is phosphorylated by Aβ$_{1-42}$ in primary cortical neurons. WT and *Fcgr2b* KO cortical neurons were incubated with 1 μM Aβ oligomers for 24 hr. Cell lysates were immunoprecipitated using FcγRIIb antibody (2.4G2) and analyzed with western blotting using phospho-tyrosine antibodies. (**C**) Inhibition of Aβ$_{1-42}$-induced FcγRIIb phosphorylation by hFcγRIIb-ED protein. SH-SY5Y cells were co-incubated with Aβ$_{1-42}$ and hFcγRIIb-ED protein. (**D**) FcγRIIb, not FcγRIIb Y273F-GFP (Y273F), is phosphorylated at Tyr273. (**E**) FcγRIIb Tyr273 phosphorylation is required for Aβ-induced tau phosphorylation. *Fcgr2b* KO neurons (DIV 8) were transfected with the indicated constructs, followed by incubation with 1 μM Aβ$_{1-42}$. Cell extracts were subjected to western blotting (*left*). The signals on the blot were quantified and bar graph represents phospho-tau levels normalized by TG5 (*right*). All data shown are means ± s.e.m.; *p<0.05, **p<0.005, one-way ANOVA. (**F**) FcγRIIb is phosphorylated at Tyr273 in AD brains. Hippocampal homogenates from normal, MCI (Braak III), and AD patients (Braak V/VI) were analyzed with western blotting (*left*). Levels of phosphorylated FcγRIIb and total FcγRIIb were quantified and values are means ± s.e.m.; **p<0.01, two-tailed *t*-test (*right*).

The following figure supplements are available for figure 3:

**Figure supplement 1.** FcγRIIb-mediated tau phosphorylation is dependent on GSK3β, not CDK5.

**Figure supplement 2.** Phosphorylation of FcγRIIb Tyr273 is required for Aβ neurotoxicity.

neurotoxicity and tau phosphorylation. Unlike FcγRIIb, ectopic expression of FcγRIIb mutants did not induce neuronal death in mouse hippocampal HT22 cells (*Figure 3—figure supplement 2A*). Moreover, overexpression of FcγRIIb Y273F mutant, which was not phosphorylated (*Figure 3D*), blocked Aβ$_{1-42}$ neurotoxicity to the control level in SH-SY5Y cells, whereas FcγRIIb WT potentiated it (*Figure 3—figure supplement 2B*), indicating that FcγRIIb-Tyr273 phosphorylation is required for Aβ$_{1-42}$ neurotoxicity. Next, we examined the role of FcγRIIb phosphorylation in tau phosphorylation using a reconstitution analysis in *Fcgr2b* KO neurons. Unlike that in *Fcgr2b*-deficient neurons, reconstitution of the *Fcgr2b* KO primary cortical neurons with FcγRIIb WT recovered Aβ-induced hyperphosphorylation of tau, detected by PHF1, CP13, AT180, and 12E8 antibodies, as seen in WT neurons (*Figure 3E*). On the other hand, reconstitution with FcγRIIb-ΔCyto or FcγRIIb-Y273F failed to show tau phosphorylation at those epitopes in response to Aβ$_{1-42}$ (*Figure 3E*). These results suggest that the phosphorylation of FcγRIIb-Tyr273 by Aβ$_{1-42}$ is critical for tau phosphorylation.

Even more interestingly, when we analyzed the phosphorylation status of FcγRIIb in the brains of AD patients, we found FcγRIIb phosphorylation on Tyr273 in the hippocampal tissues of five out of six AD patients (stage V and VI), but not in normal and mild cognitive impairment (MCI) patients (stage III) (*Figure 3F*). In addition, tau phosphorylation at PHF was observed in four out of the five AD patients with FcγRIIb-Tyr273 phosphorylation-positive brains. As we previously reported, the expression level of FcγRIIb was increased in AD brains (*Figure 3F*). With the notion that FcγRIIb phosphorylation is required for tau phosphorylation and Aβ$_{1-42}$ neurotoxicity, it is likely that the phosphorylation of FcγRIIb found in AD brains is associated with tau phosphorylation and neuronal loss during AD pathogenesis.

## Recruitment of SHIP2 to phosphorylated FcγRIIb by Aβ$_{1-42}$ in neuronal cells

In B cells, the SHIP (SHIP1 and 2) is known to bind to the phosphorylated ITIM region of FcγRIIb and inhibit downstream responses triggered by immune receptors (*Ono et al., 1996*; *Muraille, 2000*). Given that SHIP2 is highly expressed in the brain and SHIP1 is expressed predominantly in hematopoietic cells (*Astle et al., 2007*), we focused on SHIP2 and examined its ability to bind to phosphorylated FcγRIIb in neuronal cells. Immunoprecipitation assays revealed that overexpressed FcγRIIb-GFP bound to SHIP2-His in SH-SY5Y cells, whereas the phospho-defective FcγRIIb (Y273F) mutant failed to do so (*Figure 4A*). Similar results were observed in the reverse immunoprecipitation assay using GFP antibody (*Figure 4B*). Interestingly, compared to the lack of or weak interaction in untreated control cells, we found a drastic increase in the binding between endogenous SHIP2 and FcγRIIb in SH-SY5Y cells after exposure to Aβ$_{1-42}$ (*Figure 4C,D*), suggesting that SHIP2 binds to FcγRIIb in neuronal cells in response to Aβ$_{1-42}$.

Because SHIP2 is known to be localized in the cytosol but to translocate to the cell membrane upon stimulation (*Dyson et al., 2001*; *Wang et al., 2004*), we further examined subcellular localization of SHIP2 in neuronal cells exposed to Aβ$_{1-42}$. Treatment of SH-SY5Y cells with Aβ$_{1-42}$ enhanced subcellular localization of SHIP2 to the plasma membrane and thus colocalization of SHIP2 with FcγRIIb (*Figure 4E*). In addition, subcellular fractionation assays revealed similar results, showing that SHIP2 was enriched in the membrane fraction following Aβ$_{1-42}$ treatment, while it was detected in both the cytosol and membrane fractions in untreated control cells (*Figure 4F*). However, SHIP2 expression was not changed by Aβ$_{1-42}$ treatment in SH-SY5Y cells and in the brains of MCI and AD patients (*Figure 4F* and*Figure 4—figure supplement 1*). Overall, our data indicates that SHIP2 is recruited to the membrane to interact with FcγRIIb in neuronal cells following Aβ$_{1-42}$ treatment.

## Dysregulation of phosphoinositide metabolism by the FcγRIIb-SHIP2 axis for tau phosphorylation

Because SHIP2 is an inositol phosphatase that dephosphorylates PtdIns(3,4,5)P$_3$ to produce PtdIns(3,4)P$_2$ (*Damen et al., 1996*), we examined whether Aβ$_{1-42}$ affects phosphoinositide metabolism through FcγRIIb and SHIP2. We measured the levels of PtdIns(3,4)P$_2$ with a protein lipid overlay (PLO) assay using the purified His-tagged pleckstrin homology (PH) domain of tandem PH domain-containing protein-1 (TAPP1) as a probe (*Dowler et al., 2002*). From the PLO assays using total lipid extracts, we found that the amount of TAPP1-PH-interacting PtdIns(3,4)P$_2$ was increased in SH-SY5Y cells following exposure to Aβ$_{1-42}$ (*Figure 5—figure supplement 1A*). In contrast, there was no such

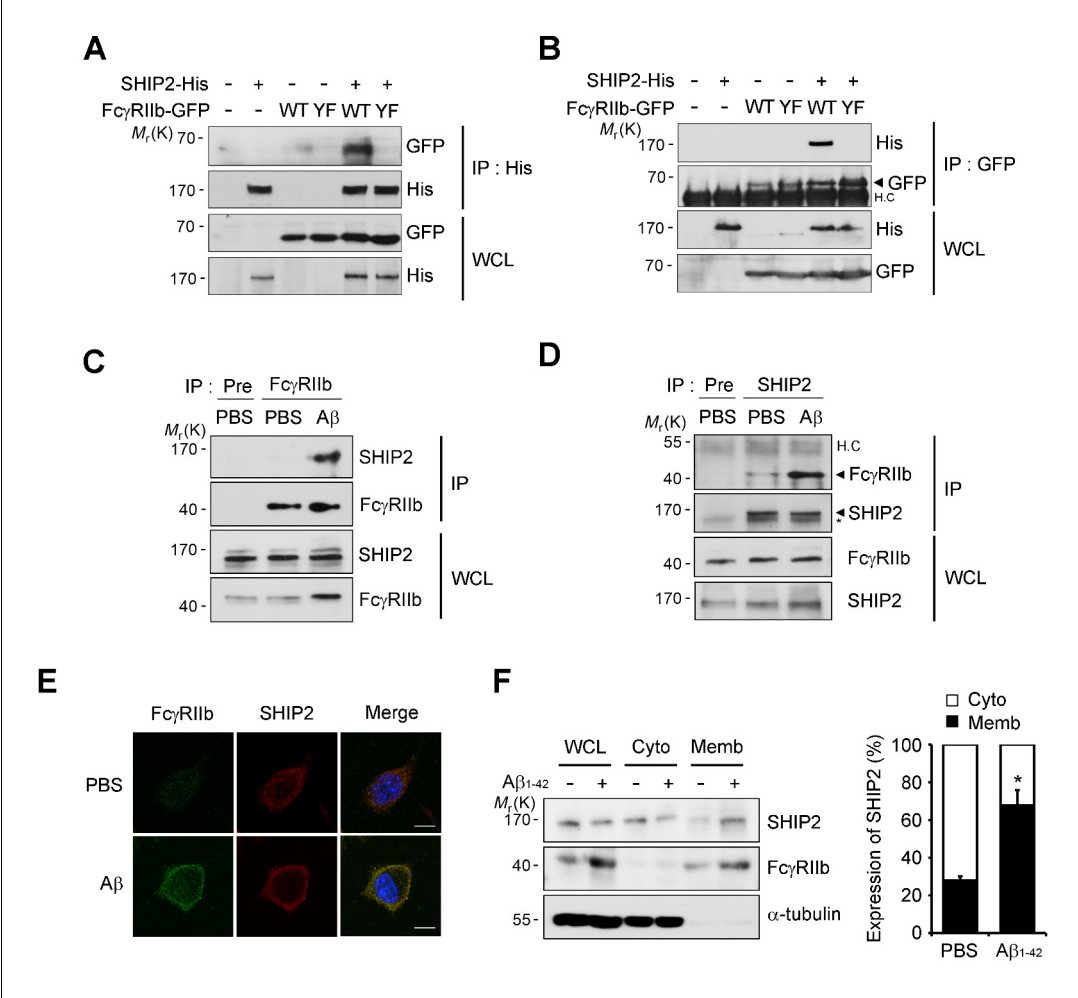

**Figure 4.** SHIP2 is recruited and binds to phosphorylated FcγRIIb in response to Aβ. (**A, B**) Interaction between phosphorylated FcγRIIb and SHIP2. SH-SY5Y cells were co-transfected with pSHIP2-His and either pFcγRIIb WT-GFP (WT) or pFcγRIIb Y273F-GFP (YF) for 36 hr and cell extracts were subjected to immunoprecipitation analysis using anti-His (**A**) and anti-GFP (**B**) antibody, respectively. Whole cell lysates (WCL) and the immunoprecipitates were probed by western blotting using anti-GFP and anti-His antibodies. (**C, D**) Regulated interaction between SHIP2 and FcγRIIb2 in response to Aβ$_{1-42}$. SH-SY5Y cells were left untreated or incubated with 1 μM Aβ$_{1-42}$ oligomers for 24 hr and cell extracts were immunoprecipitated using anti-FcγRIIb antibody (**C**) or anti-SHIP2 antibody (**D**). (**E**) Aβ induces colocalization of FcγRIIb and SHIP2 on the plasma membrane. SH-SY5Y cells were incubated with 1 μM Aβ$_{1-42}$ for 24 hr and then subjected to immunocytochemical analysis using anti-FcγRIIb and anti-SHIP2 antibodies. Hoechst dye was used for nuclear staining. (**F**) Increased targeting of SHIP2 to the plasma membrane by Aβ$_{1-42}$. SH-SY5Y cells were treated with 1 μM Aβ$_{1-42}$ for 24 hr and then subjected to subcellular fractionation assay to separate the plasma membrane from the cytosol. The fractions were analyzed by western blotting. The α-tubulin antibody was used as a marker for the cytosolic fraction (*left*). The relative expression of SHIP2 at each fraction was quantified by densitometric analysis (*right*). Values are means ± s.d.; *p<0.05, two-tailed *t*-test.

The following figure supplement is available for figure 4:

**Figure supplement 1.** Expression of SHIP2 in human brain.

increase by Aβ$_{1-42}$ in SH-SY5Y/*FCGR2B* or SH-SY5Y/*INPPL1* knockdown cells (*Figure 5—figure supplement 1B*). Conversely, the amount of the general receptor for phosphoinositides-1 (GRP1)-PH-interacting lipids, mainly PtdIns(3,4,5)P$_3$ (*Guillou et al., 2007*), was reduced by Aβ$_{1-42}$ in SH-SY5Y cells, but not in SH-SY5Y/*FCGR2B* or SH-SY5Y/*INPPL1* knockdown cells (*Figure 5—figure supplement 1B*). Similar patterns of changes in the levels of TAPP1-PH-interacting or GRP1-PH-interacting lipids were also observed in WT primary cortical neurons after treatment with Aβ$_{1-42}$ (*Figure 5A,B*).

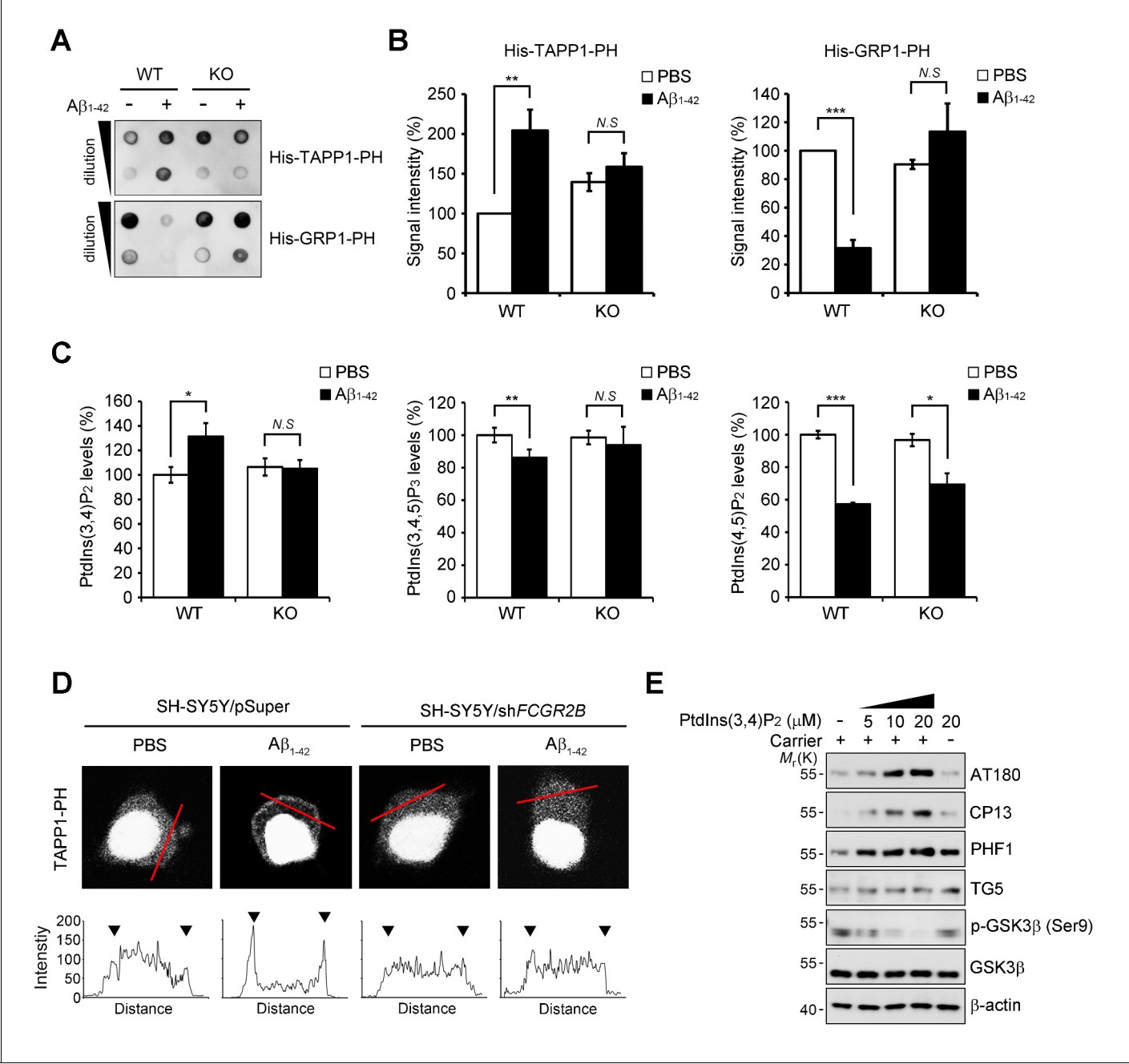

**Figure 5.** FcγRIIb-SHIP2 axis deregulates PtdIns(3,4)P$_2$ metabolism for tau phosphorylation. (A–C) Aβ$_{1-42}$ increase PtdIns(3,4)P$_2$ levels through FcγRIIb. Primary cortical neurons from WT and *Fcgr2b* KO embryos were incubated w/wo 1 μM Aβ$_{1-42}$ oligomers for 24 hr. Total lipids were extracted and analyzed by PLO assay using purified TAPP1-PH and GRP1-PH proteins which bind to PtdIns(3,4)P$_2$ and PtdIns(3,4,5)P$_3$, respectively (A) or analyzed by ELISA to quantify PtdIns levels (C). The signals on the blots in (A) were quantified by densitometric analysis (B). All data shown are means ± s.d.; **p<0.005, ***p<0.0005, unpaired *t*-test. (D) Aβ$_{1-42}$ increase PtdIns(3,4)P$_2$ levels at the plasma membrane through FcγRIIb. SH-SY5Y/pSuper and SH-SY5Y/sh*FCGR2B* stable cells were transfected with the GFP-PH$_{TAPP1}$ probe and stimulated with 1 μM Aβ$_{1-42}$ oligomers for 24 hr. The fluorescence of GFP-PH$_{TAPP1}$ was observed by confocal microscopy (*top*). The fluorescence intensity of the GFP-PH$_{TAPP1}$ probe in the plasma membrane (two external peaks) was quantified after Aβ$_{1-42}$ treatment (*bottom*). (E) Intracellular delivery of PtdIns(3,4)P$_2$ induces tau hyperphosphorylation. Primary cortical neurons were incubated with the indicated concentrations of PtdIns(3,4)P$_2$ w/wo carriers for 24 hr and cell lysates were subjected to western blotting.

The following figure supplements are available for figure 5:

**Figure supplement 1.** The FcγRIIb-SHIP2 axis is required for Aβ-induced PtdIns(3,4)P$_2$ dysregulation for tau phosphorylation.

*Figure 5 continued on next page*

*Figure 5 continued*

**Figure supplement 2.** ER stress links SHIP2 to GSK3β for tau phosphorylation.

Consistently, these changes were not observed in *Fcgr2b* KO neurons. These results indicate that phosphoinositide metabolism is affected by the FcγRIIb-SHIP2 axis.

When we directly measured the levels of phosphoinositides with an ELISA assay, we observed that the levels of PtdIns(3,4)$P_2$ were increased by 30% in primary cortical neurons after exposure to Aβ$_{1-42}$, whereas PtdIns(3,4,5)$P_3$ was decreased by 17% (*Figure 5C*). However, *Fcgr2b*-deficiency abrogated these changes in PtdIns(3,4)$P_2$ and PtdIns(3,4,5)$P_2$ levels. On the other hand, the levels of PtdIns(4,5)$P_2$ were lowered by Aβ$_{1-42}$ in WT neurons, consistent with a previous report (*Berman et al., 2008*), and also in *FcgR2b* KO neurons (*Figure 5C*). We further traced the changes in PtdIns(3,4)$P_2$ using GFP-tagged TAPP1-PH under a fluorescence microscope. While TAPP1-PH-GFP was found in a diffuse pattern in the cytosol of untreated SH-SY5Y cells, treatment with Aβ$_{1-42}$ enhanced the fluorescence of TAPP1-PH-GFP and concentrated it at the plasma membrane (*Figure 5D*). In contrast, Aβ-induced accumulation of TAPP1-PH-GFP at the plasma membrane was impaired in SH-SY5Y/*FCGR2B* knockdown cells (*Figure 5D*). As reported (*Cheung et al., 2007*), hydrogen peroxide also induced membrane localization of TAPP1-PH-GFP in SH-SY5Y cells. However, this localization was not affected by *Fcgr2b* deficiency (*Figure 5—figure supplement 1C*). These observations further support the idea that Aβ$_{1-42}$ selectively dysregulates PtdIns(3,4)$P_2$ and PtdIns(3,4,5)$P_3$ levels in neurons via FcγRIIb.

To address the important question of whether the increase in PtdIns(3,4)$P_2$ by Aβ$_{1-42}$ can influence tau phosphorylation, we directly delivered phosphoinositide into living neurons using a carrier (*Ozaki et al., 2000*). Compared to untreated control cells, treatment with PtdIns(3,4)$P_2$ increased tau phosphorylation (AT180, CP13, PHF1) in primary cortical neurons in a dose-dependent manner (*Figure 5E*). Interestingly, the increase in tau hyperphosphorylation was specific to PtdIns(3,4)$P_2$; other phosphoinositides, such as PtdIns(4,5)$P_2$, PtdIns(3,5)$P_2$, and PtdIns(3,4,5)$P_3$, failed to do so. Consistent with the activation of GSK3β by Aβ$_{1-42}$, PtdIns(3,4)$P_2$ treatment also reduced the inhibitory phosphorylation of GSK3β at Ser9 in neurons (*Figure 5—figure supplement 2A*). Moreover, PtdIns(3,4)$P_2$ induced the expression of GRP78, a typical marker of unfolded protein response (UPR), as well (*Figure 5—figure supplement 2A*). We further found that PtdIns(3,4)$P_2$-induced GSK3β activation and tau phosphorylation were attenuated by the treatment with ER stress inhibitors, such as 4-PBA and Salubrinal, a chemical chaperone and an eIF2α dephosphorylation inhibitor, respectively (*Figure 5—figure supplement 2B*). We confirmed that ER stress response and GSK3β activation triggered by Aβ were all declined by a SHIP2 inhibitor AS1949490 or lentiviral expression of *Inppl1* siRNA (*Figure 5—figure supplement 2C,D*). Combined with the previous study showing that ER stress stimulates GSK3β activity (*Ren et al., 2015*), these results suggest that an increase in the PtdIns(3,4)$P_2$ level by Aβ$_{1-42}$ activates GSK3β through ER stress for tau hyperphosphorylation in neuronal cells.

## SHIP2 is critical for Aβ-induced tau hyperphosphorylation and memory impairment

We further determined whether SHIP2, a downstream signal mediator of FcγRIIb, is essential for tau phosphorylation by examining the effects of SHIP2 knockdown. Unlike enhanced tau phosphorylation by Aβ$_{1-42}$ in control cells, knockdown of SHIP2 expression in SH-SY5Y cells abrogated tau phosphorylation by Aβ$_{1-42}$ (*Figure 6—figure supplement 1A*). Conversely, overexpression of SHIP2 alone was sufficient to increase tau hyperphosphorylation in SH-SY5Y cells (*Figure 6—figure supplement 1D*). On the other hand, these effects were not observed using an activity-dead SHIP2 mutant with Asp608 replaced by Ala (*Nakatsu et al., 2010*) (*Figure 6—figure supplement 1E*). In addition, neuronal cell death triggered by Aβ$_{1-42}$ treatment or FcγRIIb overexpression was greatly reduced by knockdown of SHIP2 expression in SH-SY5Y cells (*Figure 6—figure supplement 1B,C*). Together, these data suggest that SHIP2 is a key signal mediator of FcγRIIb in Aβ-induced tau hyperphosphorylation and neurotoxicity.

Then, we examined the role of SHIP2 (INPPL1) in memory impairment in vivo using *Inppl1* siRNA-expressing lentivirus (lenti-si*Inppl1*). We confirmed that infection with lenti-si*Inppl1* reduced SHIP2 levels in both HT22 cells and primary cortical neurons (*Figure 6—figure supplement 2A,B*). Consistently, we observed that infection with lenti-si*Inppl1* abrogated Aβ-induced tau phosphorylation (PHF1, CP13) in primary cortical neurons (*Figure 6A*). When we stereotaxically injected lenti-si*Inppl1* into the dentate gyrus of WT and 3xTg-AD mice, we observed that compared to control 3xTg AD mice, lenti-si*Inppl1*-injected 3xTg AD mice showed no significant memory deficits in Y-maze and novel object recognition tests at 20 days after viral injection (*Figure 6B,C*). When we also monitored tau phosphorylation and SHIP2 levels in brains by western blot analysis, we found that tau phosphorylation and SHIP2 levels were reduced in the hippocampi of lenti-si*Inppl1*-injected 3xTg-AD mice (*Figure 6D*). The amelioration of memory impairment by lenti-si*Inppl1* was maintained for 30 days post-injection (data not shown). These results indicate that SHIP2 is critical to memory impairment and tau hyperphosphorylation in 3xTg-AD mice.

In addition, we assessed the impact of pharmacological SHIP2 inhibition on Aβ-induced tau phosphorylation and memory impairment using a SHIP2-selective inhibitor, AS1949490, which is a 30-fold more potent inhibitor against SHIP2 than SHIP1 (*Suwa et al., 2009*). Treatment of primary cortical neurons with AS1949490 inhibited Aβ-induced tau hyperphosphorylation and neuronal cell death (*Figure 6E* and *Figure 6—figure supplement 3A*). When we examined the effect of AS1949490 on the memory impairment triggered by i.c.v.-injected $A\beta_{1-42}$ in mice, we directly delivered AS1949490 into mouse brains because of its poor bioavailability (*Suwa et al., 2010*). Behavioral tests following the i.c.v. injection of sub-lethal doses of $A\beta_{1-42}$ and AS1949490 revealed that AS1949490 significantly rescued $A\beta_{1-42}$-induced impairments of spatial working memory (*Figure 6F*), object recognition memory (*Figure 6G*), and passive avoidance memory (*Figure 6H*). Total movements, determined by arm entries in the Y-maze test, were not significantly different between the groups of mice (*Figure 6—figure supplement 3B*). Moreover, from western blot analysis of the mouse hippocampal tissues, we found that AS1949490 suppressed $A\beta_{1-42}$-induced tau phosphorylation (PHF1, CP13, AT180) (*Figure 6I*). Taken together, these observations support the view that pharmacological manipulation of SHIP2 is amenable to the development of AD therapeutics targeting $A\beta_{1-42}$-induced tau phosphorylation and memory impairment.

## Discussion

Despite tremendous efforts showing that Aβ plays a central role in the pathogenesis of AD, including memory impairment, synaptic loss, and neuronal cell death (*Cleary et al., 2005*; *LaFerla et al., 2007*), mechanistic understanding of tau phosphorylation in Aβ-induced memory deficits remains poor (*Rapoport et al., 2002*; *Roberson et al., 2007*; *Shipton et al., 2011*). To our knowledge, this is the first study showing crucial mediator, the FcγRIIb-SHIP2 axis, which is responsible for Aβ-induced tau phosphorylation, memory impairment, and neuronal loss in AD models. Because the interaction of Aβ species, especially oligomeric $A\beta_{1-42}$, with FcγRIIb accounts for such neuropathogenic defects of AD as an initiation step, the selective interaction of oligomeric $A\beta_{1-42}$ with FcγRIIb may hint at why $A\beta_{1-42}$, but not $A\beta_{1-40}$, is important in tau pathology (*Oddo et al., 2008*; *Kam et al., 2013*). Consequently, inhibition of the interaction between FcγRIIb and Aβ using anti-FcγRIIb antibody prevents Aβ-induced tau phosphorylation and memory deficits.

In general, Aβ oligomers play a key role in AD pathogenesis, while soluble Aβ oligomers are still heterogeneous, including low or high *n* oligomers, and the proposals on which species of Aβ oligomers are responsible for the pathogenesis are a little in debate (reviewed in *Benilova et al., 2012*). We have here used 3 different sources of Aβ oligomers; synthetic Aβ oligomers, naturally secreted Aβ oligomers (7PA2 cells), and Aβ of 3xTg-AD model mice. Although synthetic Aβ oligomers are well-characterized and have been used widely for neurotoxicity, the acting concentration of synthetic Aβ (μM range) is relatively higher than that in AD brains. Compared to synthetic Aβ oligomers, conditioned medium from 7PA2 cells mainly contains not only Aβ dimers and trimers but also the different pools of Aβ oligomers, including zeta peptide, and show more potent neurotoxic properties (nM range) (*Qi-Takahara, 2005*; *Haass and Selkoe, 2007*). Moreover, the oligomers generated in 3xTg-AD mouse brain may be more complicated and needs to be identified. Nonetheless, we propose here that FcγRIIb plays a crucial role in tau phosphorylation and neurotoxicity in vitro and in vivo in

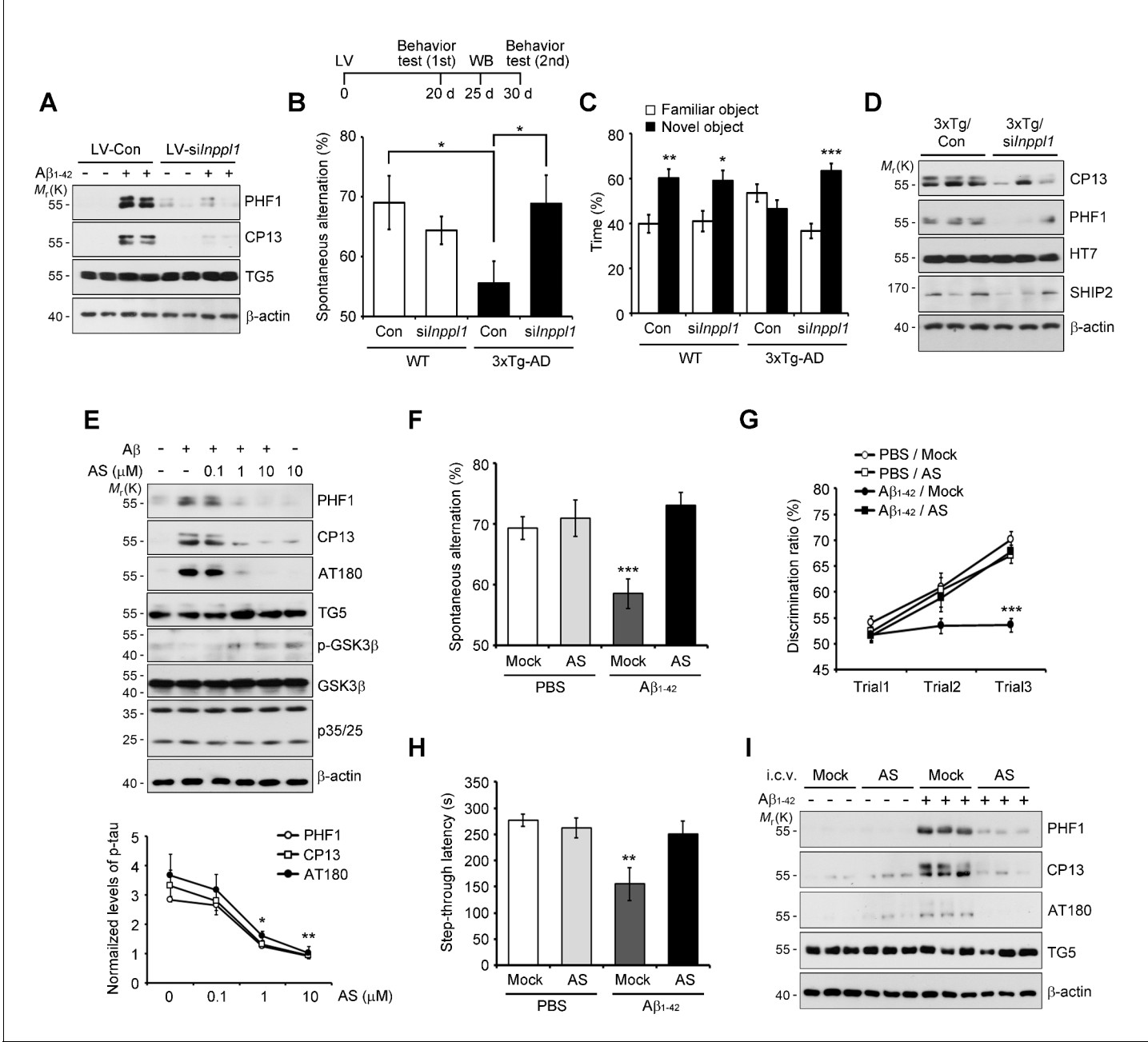

**Figure 6.** Lentiviral or pharmacological inhibition of SHIP2 prevents Aβ-mediated memory impairments and tau phosphorylation in vivo. (A) Suppression of Aβ-induced tau phosphorylation by *Inppl1* knockdown. Primary cortical neurons were infected with control (LV-Con) or *Inppl1* siRNA-containing lentivirus (LV-si *Inppl1*). On day 3, cells were incubated with 1 μM Aβ$_{1-42}$ oligomers for 24 hr and Aβ-induced tau phosphorylation was analyzed by western blotting. (B–D) Prevention of memory impairments and tau hyperphosphorylation by *Inppl1* knockdown in 3xTg-AD mice. Time schedule of the memory experiments (*top*). LV-Com or LV-si*Inppl1* was injected into the dentate gyrus of 7–8 month-old mice. Beginning 20 days after the injection, mice were subjected to Y-maze (B) and novel object recognition (C) tests (n = 7–10 mice per group). *p<0.05, **p<0.005, ***p<0.0005, one-way ANOVA. The hippocampal lysates were subjected to western blotting (D). (E) Suppression of Aβ-induced tau phosphorylation by a SHIP2 inhibitor. Primary cortical neurons were incubated with AS1949490 for 2 hr and then with 1 μM Aβ$_{1-42}$ for 24 hr. Cell extracts were analyzed with western blotting (*top*). The signals on the blots were quantified. *p<0.01, **p<0.005, unpaired *t*-test (*bottom*). (F–H) SHIP2 inhibitor prevents Aβ-induced memory deficits. The 2-month-old WT mice were injected with Aβ$_{1-42}$ alone or together with 10 μg AS1949490 (n = 10 per groups). One day later, mice were analyzed with Y-maze (F), novel object recognition (G), and passive avoidance (H) tests. **p<0.005, ***p<0.0005, one-way ANOVA. (I) Suppression of Aβ-induced tau phosphorylation in vivo by SHIP2 inhibitor. The hippocampal extracts were subjected to western blotting. All data are means ± s.e.m. (n = 3).

*Figure 6 continued on next page*

*Figure 6 continued*

The following figure supplements are available for figure 6:

**Figure supplement 1.** SHIP2 is required for Aβ-induced tau phosphorylation and neurotoxicity.
**Figure supplement 2.** Knockdown of SHIP2 expression with siRNA-carrying lentivirus
**Figure supplement 3.** A SHIP2 inhibitor prevents Aβ-induced neurotoxicity.

response to these species of Aβ, probably a certain common species among the different sources of Aβ oligomers.

Given that FcγRIIb was initially reported as a hematopoietic receptor which is mainly expressed in B cells, macrophages, and neutrophils (*Nimmerjahn and Ravetch, 2008*), our data using 3xTg-AD/ FcγRIIb KO mice raised a possibility that FcγRIIb in non-neuronal cells might contribute to APP/Aβ-induced tau pathologies via neuroinflammatory function. As expected, we observed that microglia was activated in 3xTg-AD mice and this activation was also reduced significantly in the cortex and marginally in the hippocampus by FcγRIIb deficiency (data not shown). Recently, however, we and other colleagues identified that both FcγRIIb mRNA and protein are also expressed in neurons (*Figure 1—figure supplement 1*; *Cahoy et al., 2008*; *Suemitsu et al., 2010*; *Kam et al., 2013*), though FcγRIIb expression in neurons is low compared to astrocytes and immune tissues. Notably, its expression is increased at least several folds in primary neurons after exposure to $Aβ_{1-42}$ and in AD brain (*Kam et al., 2013*). In addition, transgenic expression of the FcγRIIb mutant lacking its cytoplasmic domain in the neurons blocked memory impairment in 3xTg AD mice (data not shown). Moreover, the results showing that Aβ-induced tau phosphorylation was prevented by FcγRIIb deficiency, antagonistic FcγRIIb antibody, or SHIP2 inhibitor in primary cultured neurons assure neuronal function of the FcγRIIb-SHIP2 axis. Together, we believe that the inflammation mediated by FcγRIIb in 3xTg-AD mice is not likely a major cause of the memory impairment but contributes to the aggravation of memory impairment in the mice.

Unlike other FcγRs that have an immunoreceptor tyrosine-based activating motif (ITAM), FcγRIIb has a unique ITIM in its cytoplasmic tail and thus acts as an inhibitory receptor in B cells (*Okun et al., 2010*). Excitingly, the binding of extracellular $Aβ_{1-42}$ to FcγRIIb induces phosphorylation in the ITIM of FcγRIIb in neuronal cells. We observed that Lyn kinase is expressed in neurons and that knockdown of *LYN* expression blocks Aβ-induced FcγRIIb phosphorylation and neurotoxicity (data not shown). Thus, it is likely that Lyn phosphorylates the ITIM of FcγRIIb in neuronal cells in response to $Aβ_{1-42}$. Moreover, we also observed this phosphorylation of FcγRIIb in AD brains (stage V and VI) in which tau was highly phosphorylated. Then, how is FcγRIIb different from other Aβ receptors? It is reasonable to propose that different mechanisms or even the same mechanism exerts multiple effects at different stages of disease progression (*De Strooper and Karran, 2016*). For instance, RAGE is now believed to mainly function to transport Aβ in the blood brain barrier (*Deane et al., 2003*) and ABAD acts for mitochondrial toxicity as an intracellular binding partner of Aβ (*Lustbader, 2004*). In case of PrPc, it's debatable whether it is involved in Aβ-induced memory impairments and thus needs to be further characterized (*Balducci et al., 2010*; *Gimbel et al., 2010*; *Cissé et al., 2011b*). Further, compared to those receptors, our observations that the phosphorylation of FcγRIIb at tyrosine 273 is found in the brain of AD patients and is required for both oligomeric Aβ neurotoxicity and tau hyperphosphorylation can make it distinct from other Aβ-binding receptors. In the case of PirB that shares structural similarity with FcγRIIb and also acts as an Aβ receptor for synaptic plasticity, the phosphorylation of PirB is not associated with Aβ signaling (*Kim et al., 2013*). Thus, we believe that FcγRIIb facilitates tau phosphorylation and neuronal loss in AD brains, consistent with the proposed role of tau in AD pathogenesis, such as severe memory impairment and neuronal loss (*Ballatore et al., 2007*).

Interestingly, we show that SHIP2 is a key mediator in delivering the toxic signal of $Aβ_{1-42}$ to tau by binding to the phosphorylated FcγRIIb. Many studies on SHIP2 have focused on its inhibitory effect on insulin signaling (*Ishihara et al., 1999*; *Wada et al., 2001*). *Inppl1* transgenic mice show impaired insulin signaling and glucose intolerance, while *Inppl1* KO mice are highly resistant to

weight gain on a high-fat diet (*Sleeman et al., 2005*; *Kagawa et al., 2008*). Thus, targeting SHIP2 is thought to be a promising approach for the treatment of other diseases, including type 2 diabetes (*Vanhanen et al., 2006*). Given that metabolic syndrome, including insulin resistance and glucose intolerance, is highly associated with AD (*Vanhanen et al., 2006*; *Razay et al., 2007*), we speculate that SHIP2 may play a dual role in AD and diabetes. In support of this, a SHIP2 inhibitor exhibited an ameliorating effect on the impaired memory function of diabetic mice (*Soeda et al., 2010*). Several single-nucleotide polymorphisms (SNPs) of SHIP2 are involved in metabolic syndrome (*Kaisaki et al., 2004*; *Kagawa et al., 2005*); thus, it will be very interesting to evaluate the effect of these SHIP2 SNPs on AD pathogenesis in our FcγRIIb-SHIP2 axis.

Dysregulation of phosphoinositide metabolism is increasingly recognized as important in various diseases, such as cancer, diabetes, and myopathy (*Wymann and Schneiter, 2008*; *Kok et al., 2009*). The phosphoinositide pool is also altered in AD (*Stokes and Hawthorne, 1987*; *Jope et al., 1994*). In particular, this work shows that a change in the PtdIns(3,4)$P_2$ level is implicated in AD pathogenesis through tau hyperphosphorylation. PtdIns(3,4)$P_2$ is scarce under normal conditions and increases through signaling (*Lemmon, 2008*). While PtdIns(3,4)$P_2$ and PtdIns(3,4,5)$P_3$ are known to overlap to some degree in their functions, they apparently have distinct roles in neurodegeneration. Recently, it was shown that PtdIns(3,4)$P_2$, but not PtdIns(3,4,5)$P_3$ and PtdIns(4,5)$P_2$, potentiates glutamate-induced cell death in neurons (*Sasaki et al., 2010*). In addition, PtdIns(3,4)$P_2$ phosphatase INPP4A deficiency shows increased level of PtdIns(3,4)$P_2$ and leads to neurodegeneration in brains (*Sasaki et al., 2010*). We also observed that PtdIns(3,4)$P_2$ selectively induces tau hyperphosphorylation in neurons. In addition, tensin homolog deletion on chromosome 10 (PTEN) also dephosphorylates PtdIns(3,4,5)$P_3$ at position 3 and generates PtdIns(4,5)$P_2$ (*Li et al., 1997*). The reduced level of PTEN in AD brains correlates with an increase in tau phosphorylation and, thus, dysregulation of PTEN also contributes to tau pathology (*Kerr et al., 2006*; *Zhang et al., 2006*). More recently, genetic reduction of synaptojanin 1 (Synj1), the major PtdIns(4,5)$P_2$ phosphatase in the brain, ameliorates behavioral and synaptic deficits and accelerates Aβ clearance through rescuing PtdIns(4,5)$P_2$ deficiency (*McIntire et al., 2012*; *Zhu et al., 2015*). Because oligomeric Aβ decreases PtdIns(4,5)$P_2$ levels in AD (*Berman et al., 2008*), as we also observed, there is a possibility that SHIP2 activation together with either Synj1 upregulation or PTEN downregulation induces the imbalance in the phosphoinositide pool between PtdIns(3,4)$P_2$ and PtdIns(4,5)$P_2$, and leads to AD pathogenesis.

In conclusion, the FcγRIIb-SHIP2 signaling axis provides the missing link between Aβ and tau pathologies. Notably, Aβ$_{1-42}$ induces FcγRIIb phosphorylation to recruit SHIP2, leading to disruption of phosphoinositide metabolism for tau hyperphosphorylation and memory impairment in neurons and AD model mice. Together with the Aβ-lowering strategy, our results provide new ways for AD therapeutics to rescue Aβ and tau pathology: i) selective inhibition of the interaction between FcγRIIb and Aβ$_{1-42}$, ii) inhibition of a kinase (i.e., Lyn) which phosphorylates FcγRIIb upon Aβ$_{1-42}$ stimulation, and iii) inhibition of SHIP2 which disrupts phosphoinositide metabolism.

## Materials and methods

### FcγRII b knockout, 3xTg-AD and hAPP (J20) transgenic mice and tissue preparation

WT (C57BL/6), *Fcgr2b* KO C57BL/6 (*Takai et al., 1996*), 3xTg-AD and hAPP (J20, The Jackson Laboratory, Bar Harbor, ME) mice were used. All experiments involving animals were performed according to the protocols approved by the Seoul National University Institutional Animal Care and Use Committee (SNU IACUC) guidelines. For biochemical assays, mice were anesthetized and the brains were rapidly dissected into subregions (cortex and hippocampus) and snap-frozen at –80℃. The hemibrains were homogenized in lysis buffer [20 mM Tris-HCl (pH 7.4), 150 mM NaCl, 1% Triton X-100, protease inhibitor cocktail] and centrifuged at 14,000 *g* for 20 min at 4℃. Supernatant was then collected and the protein concentration was determined using the Bradford method (GE Healthcare).

### Preparation of human brain samples

Hippocampal tissues from AD (Braak V-VI) (mean age, 80.2 ± 10.6 years; mean post-mortem intervals, 16.3 ± 6.7 hr), MCI (Braak III) (mean age, 84.2 ± 2.9 years; mean post-mortem intervals,

20.1 ± 8.0 hr) and non-AD patients (mean age, 67.8 ± 16.5 years; mean post-mortem intervals, 20.1 ± 5.8 hr) were kindly provided by the Harvard Brain Tissue Resource Center (McLean Hospital). Hippocampal tissues of AD patients were homogenized in ice-cold Tris-buffered saline (TBS) buffer [20 mM Tris-HCl (pH 7.4), 150 mM NaCl and protease inhibitor cocktails]. The homogenates were clarified by centrifugation at 14,000 $g$ for 20 min at 4°C, aliquoted and stored at –80°C until use. The supernatants were subjected to SDS-PAGE.

## Stereotaxic and intracerebroventricular injection, and lentivirus

Seven to 8-month-old WT and 3xTg-AD mice were deeply anesthetized with a mixture of ketamine (100 mg/kg) and xylazine (10 mg/kg). Lentivirus expressing shRNA against mouse *Inppl1* (5′-GAA GGG AGG GCA CGT TAA TTT-3′) (Sigma-Aldrich) was used for injection and pLKO.1-Neo-CMV-tGFP non-target virus was used as a control. The lentivirus ($1.1 \times 10^9$ TU/ml, TU; transduction unit) was stereotaxically injected bilaterally into the dentate gyrus (2 µl per hemisphere at 0.4 µl/min) with the following coordinates: anteroposterior = 2.1 mm from bregma, mediolateral = ± 1.8 mm, dorso-ventral = 2.0 mm. After the injection, the cannula was maintained for an additional 5 min for a complete absorption of the virus. Behavior tests were performed 20 and 30 days after the injection. For western blot analysis, brains were removed 25 days after viral injection, hippocampal region was dissected and its protein samples were prepared as described above. Intracerebroventricular injection of PBS or Aβ$_{1-42}$ (Sigma-Aldrich, St. Louis, MO) was performed as described previously (*Kam et al., 2013*). In the case of coinjection experiments, oligomeric Aβ$_{1-42}$ was incubated alone or together with AS1949490, IgG or 2.4G2 antibody before use.

## Behavior tests

Behavior tests for double transgenic or injected mice were performed as described previously (*Kam et al., 2013*). All apparatus and objects were cleaned with 70% ethanol before and after each trial. In Y-maze test, the mice were placed in the end of one arm (32.5 cm length × 15 cm height) of apparatus and allowed to move freely for 7 min. When the all four paws were into the arm, the entry was counted. The percentage of spontaneous alternations was calculated as the ratio of the number of successful alternations to the number of total alternations. In novel object recognition test, the mice were habituated in a chamber (22 cm wide × 27 cm long × 30 cm high) for 7 min with 24 hr intervals. In training trial #1 (2 days after habituation), the mice were exposed to two objects and allowed to explore freely for 7 min. In the testing trial #2 (a day after training), one of the familiar objects was replaced to a novel object and recognition was counted for 7 min. The same test was repeated after 24 hr, but with another novel object (trial #3). The object recognition was defined as spending time with orienting toward the object in a distance of 1 cm or less, sniffing the object or touching with the nose. The passive avoidance test was done in an apparatus consist of a light and dark compartment (20 × 20 × 20 cm each) separated by a guillotine door. The mice were allowed to explore the box for 5 min with the door open for habituation, and then returned to home cage. After 24 hr, the mice were placed into the light compartment for conditioning. The door was closed after entering into the dark room and then an electric foot shock (0.25 mA, 2 s) was delivered by the floor grids. The latency time for mice to enter the dark room was measured with a 5 min cut-off after 24 hr.

## Synthetic and naturally secreted Aβ oligomers

Synthetic Aβ$_{1-42}$ oligomers were prepared from lyophilized monomers (rPeptide, Bogart, GA). The hydroxyfluroisopropanol (HFIP)-treated Aβ$_{1-42}$ peptide was dissolved in dimethylsulfoxide (DMSO) and then diluted in phosphate-buffered saline (PBS). The stock solution was incubated at 4°C for 24 hr and stored at –80°C until use. Before use, the solution was centrifuged at 12,000 $g$ for 10 min and the supernatant was used as an oligomeric Aβ (ADDLs). The oligomeric status of Aβ$_{1-42}$ was evaluated by western blot analysis and atomic force microscopy (*Kam et al., 2013*). The conditioned medium of 7PA2-CHO cells (kindly provided by Dr. D.J. Selkoe, Harvard Medical School) were collected and used as a naturally secreted Aβ oligomers.

## ELISA analysis for Aβ quantification

Aβ levels in the 3xTg-AD and 3xTg-AD/*Fcgr2b* KO mice were analyzed using a sandwich enzyme-linked immunosorbent assay (ELISA) kit (Invitrogen, Carlsbad, CA) following the manufacturer's instructions. Briefly, the mice were anesthetized and the brains were microdissected. The hippocampus was carefully isolated and homogenized in 10 volumes of ice-cold guanidine buffer (5 M guanidine-HCl/50 mM Tris-Cl, pH 8.0) and then mixed for 3 hr at room temperature. The brain homogenates were further diluted 1:10 with cold reaction buffer (5% BSA, 0.03% Tween-20, and 5 mM EDTA in PBS supplemented with protease inhibitor cocktail) and then centrifuged at 16,000 $g$ for 20 min at 4°C. The diluents were mixed 1:1 with standard dilution buffer in the assay kit.

## Cell culture, DNA transfection and treatments

Primary cortical and hippocampal neurons were cultured from embryonic day 17 (E17) mice. The neurons were plated on poly-L-lysine (0.01% in 100 mM borate buffer, pH 8.5)-coated glass coverslips and maintained in neurobasal medium containing 2% B-27 supplement (Invitrogen) and 0.5 mM L-glutamine (Invitrogen). Half of the medium was exchanged every 3 days. Primary neurons were not authenticated, and were not tested for mycoplasma. SH-SY5Y, HEK293T and CHO cells (ATCC, Manassas, VA), which were authenticated by ATCC and were negative for mycoplasma, were cultured in DMEM (HyClone, Logan, UT) supplemented with 10% fetal bovine serum (FBS) (HyClone), penicillin and streptomycin (Invitrogen). Cells were grown at 37°C under an atmosphere of 5% $CO_2$. Primary neurons were transfected using Lipofectamine 2000 reagent (Invitrogen), whereas other cells were transfected using Polyfect reagent (Qiagen, Germany) according to the manufacturer's instructions. If required, cells were treated with SB-415286, roscovitine (Sigma-Aldrich) or AS1949490 (Tocris, United Kingdom) as indicated.

## DNA constructs

All primers used in this study are listed in *Supplementary file 1*. Human FCGR2B cDNAs was amplified by PCR from a human brain cDNA library and subcloned into pEGFP-N1 vector. The cDNAs of FCGR2B deletion mutants, ΔCyto, ΔITIM and ΔC-term was generated by PCR and subcloned into pEGFP-N1 vector. A point mutant of FCGR2B (Y273F) was generated by site-directed mutagenesis. All mutants were confirmed by DNA sequencing analysis. Human *FCGR2B* and human *INPPL1* shRNAs were synthesized, annealed and cloned into the pSUPER-neo vector. Human tau (0N4R) cDNAs were subcloned into pcDNA3-HA and pEGFP-C1 vector as described previously (*Park et al., 2012*). His-tagged mouse Ship2 cDNA is a generous gift from Dr. M.G. Tomlinson (University of Birmingham, UK). GFP-tagged mouse Ship2 and D608A mutant were kindly provided by Dr. P. De Camilli (Yale University).

## Generation of stable cell lines

SH-SY5Y cell were transfected with pcDNA3-HA, pFCGR2B-HA, pSuper-neo, p*FCGR2B* shRNAs or p*INPPL1* shRNAs for 36 hr and then cultivated in the selection medium containing 1 mg/ml G418 (Invitrogen) for at least two weeks. A single cell was further cultivated to form stable cell colony and the expression of each cell lines was analyzed by western blotting and reverse transcriptase (RT) PCR.

## Western blot analysis and antibodies

Cells were lyzed in lysis buffer (50 mM Tris-HCl pH 7.4, 30 mM NaCl, 1% Triton X-100, 0.1% SDS, 1 mM EDTA, 1 mM PMSF, 1 mM $Na_3VO_4$, 1 mM NaF, 1 μg/ml each of aprotinin, leupeptin and pepstatin A). The lysates were centrifuged at 14,000 g for 10 min at 4°C and the supernatant was separated by SDS-PAGE and blotted onto PVDF membrane. The blots were blocked for 1 hr at room temperature and incubated with following antibodies: anti-phospho-FcγRIIb, anti-FcγRIIb (Epitomics, Burlingame, CA), anti-Aβ (4G8, Signet, Dedham, MA), anti-phospho-tau (AT180 and AT100, Innogenetics, Alpharetta, GA), anti-NSE (Zymed, South San Francisco, CA), anti-phospho-GSK3β, anti-p35/25 (Cell signaling, Danvers, MA), anti-GSK3β (BD Biosciences, San Jose, CA), anti-SHIP2, anti-mFcγRIIb, anti-GFP, anti-His (Santa Cruz Biotechnology Inc., Dallas, TX), anti-α-tubulin and anti-β-actin (Sigma-Aldrich), Nu-1 (kindly provided by Dr. W.L. Klein, Northwestern University), PHF1, CP13 and TG5 (a generous gifts from Dr. P. Davies, Albert Einstein College of Medicine), and

12E8 (a generous gift from Dr. P. Seubert (Elan Pharmaceuticals). Membranes were rinsed three times with TBS-T (10 mM Tris-Cl, pH 7.5, 150 mM NaCl, 0.1% Tween-20), further incubated for 1 hr with peroxidase-conjugated secondary antibodies and visualized using ECL detection system.

## Immunocytochemistry

Mouse primary cortical or hippocampal neurons were fixed in 4% paraformaldehyde (PFA) (Sigma-Aldrich) for 10 min, rinsed three times with PBS and then permeabilized with 0.1% Triton X-100 in PBS. After blocking with 5% BSA in PBS, neurons were incubated overnight at 4°C with the following antibodies: AT180 (1:200), AT8 (1:200), anti-NSE (1:200), anti-FcγRIIb (1:500) and anti-SHIP2 (1:250). After rinsing three times with PBS, cells were incubated with FITC- or TRITC-conjugated secondary antibodies (Jackson Laboratory Inc.) at room temperature for 1 hr. The coverslips were placed with mounting solution (Sigma-Aldrich) and observed on a confocal laser scanning microscope (Carl Zeiss Inc., Thornwood, NY).

## Immunoprecipation assay

For endogenous immunoprecipitation (IP) assay, $A\beta_{1-42}$-treated SH-SY5Y cell extracts were incubated with anti-FcγRIIb or anti-SHIP2 antibodies in IP buffer [50 mM Tris-Cl (pH7.4), 150 mM NaCl, 1% Triton X-100, 1 mM EDTA, a mixture of protease inhibitors] for 12 hr at 4°C, and then pulled-down by protein G-Sepharose beads (GE Healthcare, United Kingdom). For co-immunoprecipitation assay, HEK293T cells which transiently overexpressed His-tagged SHIP2 with either GFP-tagged FcγRIIb or FcγRIIb Y273F mutant were lyzed and incubated with anti-GFP or anti-His antibodies for 12 hr at 4°C, and then pulled-down by protein G-Sepharose beads. After a short centrifugation, the beads were washed three times with IP buffer and subjected to western blotting.

## Subcellular fractionation

SH-SY5Y cells treated with $A\beta_{1-42}$ were harvested with the buffer (20 mM Tris-HCl pH7.5, 150 mM NaCl, 0.1% Triton X-100, 1 mM EDTA) and then mechanically disrupted using a 26-gauge needle with passing it 20 times. Cell lysates were centrifuged at 1000 g for 10 min at 4°C to remove the nuclei or unbroken cells. The supernatant was collected into new tube and again centrifuged at 100,000 g for 1 hr at 4°C with Beckman SW41 rotor. The pellet was resuspended in lysis buffer and used as a crude membrane fraction, whereas the supernatant used as a cytosolic fraction. The separated fractions were confirmed with western blot analysis using anti-FcγRIIb and anti-α-tubulin antibodies for the membrane and cytosolic markers, respectively.

## Lipid extraction

The lipid extraction from primary cortical neurons or SH-SY5Y cells was performed as described (*Gray et al., 2003*). After stimulation, cells were harvested with ice-cold 0.5 M trichloroacetic acid (TCA) solution, standing on ice for 5 min and centrifuged at 200 g for 5 min. The pellet was washed with 5% TCA with 1 mM EDTA solution. Neutral lipids were extracted from the pellet with a 2:1 solution of methanol and chloroform, followed by vigorous vortexing for 10 min at room temperature. The extracts were centrifuged at 200 g for 5 min, and the acidic lipids were then extracted. A 80:40:1 solution of methanol, chloroform and 12 M HCl was added to the pellet and vortexed for 15 min at room temperature, and then centrifuged at 200 g for 5 min. The 750 µl of supernatant was transferred to a new tube and added with 250 µl of chloroform and 450 µl of 0.1 M HCl. After mixing, the samples were centrifuged to separate the organic and aqueous phases and the lower organic phase was collected and dried under vacuum. The lipids were then resuspended by sonication in a water bath with an appropriate buffer (1:2:0.8 solution of chloroform, methanol and water for PLO assay, or PBS-T buffer for ELISA).

## Protein lipid overlay (PLO) assay and ELISA

For the purification of recombinant proteins, the PH domain of TAPP1 and GRP1 was inserted into pET-28a vector. *E.coli* BL21 cells were transformed with the plasmids and cultured to reach an $OD_{600}$ of 0.6, before induction with 1 mM IPTG. After incubation for 18 hr at 16°C, cells were harvested and lyzed by sonication. His-fused TAPP1-PH and GRP1-PH were purified from cell lysates using Ni-NTA chelating agarose CL-6B (Peptron, Korea). PLO assays were performed as described

previously with minor modification (*Dowler et al., 2002*). Lyophilized PtdIns(3,4)$P_2$, PtdIns(4,5)$P_2$ and PtdIns(3,4,5)$P_3$ diC16 (Echelon, Salt Lake City, UT) were reconstituted in a 1:2:0.8 solution of chloroform, methanol and water, and used as positive controls. The lipid extracts from the cells were serially diluted and spotted on PVDF membranes which were pre-wetted in methanol, washed in TBS-T buffer and then air-dried. The membranes were dried completely and blocked with 3% BSA in TBS-T buffer for 1 hr. The blots were incubated overnight at 4°C with gentle rocking in the fresh blocking buffer containing purified 10 μM His-TAPP1-PH or His-GRP1-PH. The membranes were washed 5 times over 50 min in TBS-T buffer and incubated with anti-His antibody for 1 hr at room temperature. The membranes were further incubated for 1 hr with peroxidase-conjugated secondary antibodies and bound proteins were visualized using ECL detection system. PtdIns(3,4)$P_2$, PtdIns(4,5)$P_2$ or PtdIns(3,4,5)$P_3$ levels were quantified by ELISA kit (Echelon) following the manufacturer's instructions.

### Intracellular delivery of phosphoinositide

The synthetic phosphoinositides diC16 were incubated with histone carriers (Echelon) with a 0.5–3:1 molar ratio for 15 min with a vigorous vortexing. The histone-phosphoinositides complex was diluted 1:10 with neurobasal media and added to primary cortical neurons. The phosphoinositides without carriers and the only carriers without phosphoinositides were used as negative controls.

### Statistical analyses

Statistical analyses were performed with GraphPad Prism software. Differences between two means were assessed by paired or unpaired *t*-test. Differences among multiple means were assessed by one-way ANOVA, followed by Tukey's post-hoc test. Error bars represent s.d. or s.e.m. as indicated.

## Acknowledgements

The authors thank Dr. U Hammerling (Memorial Sloan Kettering Cancer Center, NY) for FcγRIIb monoclonal antibody (K9.361 hybridoma), Dr. J Cambier (University of Colorado Health Sciences Center, CO) for FcγRIIb rabbit polyclonal antibody, Dr. DJ Selkoe (Harvard Medical School, MA) for CHO and 7PA2 cells, Dr. WL Klein (Northwestern University, IL) for oligomeric Aβ antibody, Dr. P Davies (Albert Einstein College of Medicine, NY) for PHF1, CP13, and TG5 antibodies, Dr. P Seubert (Elan Pharmaceuticals, CA) for 12E8 antibody, Dr. MG Tomlinson (University of Birmingham, Birmingham, UK) for SHIP1 and SHIP2 cDNAs, Dr. P De Camilli (Yale University, CT) for SHIP2 D608A cDNA. YD Gwon, SH Kim, and SW Moon were in part supported by the BK21 program and Global Ph.D. program. AD tissues were provided from the Harvard Brain Tissue Resource Center of McLean Hospital, MA. This work was support by the CRI grant (NRF-2016R1A2A1A05005304), the National Research Council of Science & Technology (NST) grant (MSIP Project No. 2N41660-16-074) and Global Research Laboratory (NRF-2010–00341) funded by the Ministry of Education, Science, and Technology.

## Additional information

### Funding

| Funder | Grant reference number | Author |
| --- | --- | --- |
| National Research Foundation of Korea | CRI grant, NRF-2016R1A2A1A05005304 | Tae-In Kam Hyejin Park Youngdae Gwon Seo-Hyun Kim Yong-Keun Jung |
| National Research Council of Science and Technology | MSIP Project No. 2N41660-16-074 | Youngdae Gwon |
| National Research Foundation of Korea | Global Research Laboratory, NRF-2010-00341 | Sungmin Song Seo Won Moon Yong-Keun Jung |

The funders had no role in study design, data collection and interpretation, or the decision to submit the work for publication.

## Author contributions

T-IK, HP, Conception and design, Acquisition of data, Analysis and interpretation of data, Drafting or revising the article, Contributed unpublished essential data or reagents; YG, Acquisition of data, Drafting or revising the article, Contributed unpublished essential data or reagents; SS, Analysis and interpretation of data, Contributed unpublished essential data or reagents; S-HK, SWM, Acquisition of data, Contributed unpublished essential data or reagents; D-GJ, provided the 3xTg-AD mice and technical assistance., Contributed unpublished essential data or reagents; Y-KJ, Conceived and directed the project, Conception and design, Analysis and interpretation of data, Drafting or revising the article, Contributed unpublished essential data or reagents

## Author ORCIDs

Yong-Keun Jung, http://orcid.org/0000-0002-9686-3120

## Ethics

Human subjects: All experiments using human brain samples were performed according to the protocol approved by the Seoul National University Institutional Review Board (SNU IRB, permit number: E1212/001-006).

Animal experimentation: All experiments involving animals were performed according to the protocols approved by the Seoul National University Institutional Animal Care and Use Committee (SNU IACUC) guidelines (Permit Number: SNU-130722-5-2).

## Additional files

**Supplementary files**
• Supplementary file 1. Primer sequences for cloning, shRNA and qPCR.

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
