## [Decision Letter]

Thank you for submitting your article "FcγRIIb-SHIP2 axis links Aβ to tau pathology by disrupting phosphoinositide metabolism in Alzheimer's disease model" for consideration by *eLife*. Your article has been reviewed by three peer reviewers, and the evaluation has been overseen by a Reviewing Editor and a Senior Editor.

The work describes how FcγRIIb deficiency protects neurons against Aβ-induced toxicity. Especially the identification of SHIP2 linking Aβ to Tau pathology is novel and potentially of high interest. Overall the referees were in support of publication, but before reaching consensus a few critical questions should be addressed. The reviewers have discussed the reviews with one another and the Reviewing Editor has drafted this decision to help you prepare a revised submission.

Essential revisions:

1) Overall the paper should take more into account other publications in both Introduction and Discussion sections. There are in the main time at least ten other receptors for Aβ oligomers described. Also some critical perspective on what is meant with Aβ oligomers is indicated. For instance, the 7PA2 cell line also produces 'zeta' peptides and the relation between these in vitro generated mixtures of Aβ peptides and the in vivo situation is far from clear. It is important that the broader non-specialized reader of *eLife* is not misled in the sense that this paper resolves the controversies in the field. Therefore, critical literature should be included in the Introduction and Discussion, and some balancing of the conclusions in the current manuscript versus other work should be provided. Why do the authors think that their receptor is the major driver of pathology in AD and why are other receptors not considered? Does this need any qualifiers?

2) There are two major gaps in the proposed cascade and this should be mentioned at least in the Discussion to clarify this for the reader. First there are no mechanistic data in this work to link Aβ to FcγRIIb phosphorylation. The authors mention lyn kinase briefly (Discussion, third paragraph) but don't show any data. The second gap is a mechanistic link between increased PtdIns(3,4) and tau pathology. GSK-3-β kinase that catalyzes tau hyperphosphorylation, uses ATP as a donor of phosphate groups, so phosphoinositides are not a substrate. Therefore a possible link could be PtdIns(3,4)-mediated regulation of GSK-3β.

3) Two referees have concerns with the idea that the action of FcγRIIb is cell autonomous. First there are issues with the expression of FcγRIIb in neurons: Figure.1——figure supplement 1 A shows in fact that a large part of FcγRIIb mRNA is in glia and not in neurons. Additional evidence to proof that FcγRIIb is expressed in neurons is needed, it could be that the weak signal in the experiments in neurons is background. The authors could consider in situ hybridisation to check their claim. In addition, FcγRIIb is highly expressed in microglial cells and may play a role in microglial activation. Therefore, the impact of FcγRIIb KO on tau pathology in 3X TG mice can be also mediated by reduced microglial activation (cell autonomous effects vs non-autonomous effects). Did the authors check microglial activation and cytokine levels in these mice (WT 3XTG vs. KO 3Xtg)? Moreover, have the authors looked at whether FcγRIIb plays a role in sensing Aβ in microglial cells?

4) The authors should provide in the next version the pictures of Figure 1—figure supplement 2 which are now missing. These data are quite important as demonstrating the effect of the FcγRIIb receptor deficiency on APP processing.

5) In the last paragraph of the subsection “FcγRIIb is essential for tau hyperphosphorylation and memory deficit in 3xTg-AD mice”, it is said that 9months old 3xTgAD mouse are "plaque-free". What happens in the cortex of these 3xTg mice where Aβ plaques should be present, is tau phosphorylation more pronounced there? And is that protected by FcγRIIb deficiency? Does the age of mice affect tau hyperphoshorylation, i.e. is tau pathology still absent in 3xTg FcγRIIb KO mouse even at a very advanced age when there are loads of Aβ?

6) In Figure.3—figure supplement.2A the authors analyze the effects of the receptor mutations on Aβ-induced cell death. "Cell death was determined after 24h under fluorescence microscope". What method/dye/criterion was used? How many spots were analyzed per sample per condition to produce statistics on panel B?

---

## [Author Response]

[…] Essential revisions:

*1) Overall the paper should take more into account other publications in both Introduction and Discussion sections. There are in the main time at least ten other receptors for Aβ oligomers described. Also some critical perspective on what is meant with Aβ oligomers is indicated. For instance, the 7PA2 cell line also produces 'zeta' peptides and the relation between these in vitro generated mixtures of Aβ peptides and the in vivo situation is far from clear. It is important that the broader non specialized reader of eLife is not misled in the sense that this paper resolves the controversies in the field. Therefore, critical literature should be included in the Introduction and Discussion, and some balancing of the conclusions in the current manuscript versus other work should be provided. Why do the authors think that their receptor is the major driver of pathology in AD and why are other receptors not considered? Does this need any qualifiers?*

Until now, Aβ was reported to bind to many receptors, including alpha7 nicotinic acetylcholine receptors (α7 nAChR), NMDA receptor, receptors for advanced glycation end-products (RAGE), Aβ-binding alcohol dehydrogenase (ABAD), the Ephrin-type B2 receptor (EphB2), cellular prion protein (PrPc) and paired immunoglobulin-like receptor B (PirB) (Yan et al., 1996; Wang et al., 2000; Lustbader et al., 2004; Snyder et al., 2005; Lauren et al., 2009; Cisse et al., 2011a; Kim et al., 2013). Although these receptors were shown to be responsible for Aβ neurotoxicity, especially memory impairment in AD mice, their role as neuronal receptors in Aβ-induced tau pathologies was limitedly shown in α7 nAChR and NMDA receptor (reviewed in Stancu et al., 2014). Of particular note, while α7 nAChR was reported to mediate Aβ-induced tau phosphorylation, the finding was based on in vitroand ex vivosystem (Wang et al., 2003). It thus remains to be confirmed in detail using animal models of AD. Furthermore, evidence showing a correlation of the proposed molecular mechanism with pathologic evidence was not much provided. In particular, the CAMKK2-AMPK at down-stream of NMDA receptor was recently proposed to mediate the synaptotoxic effects of Aβ oligomers through tau phosphorylation and this event is very likely caused by NMDA receptor-induced increase of intracellular calcium, not by direct interaction of NMDA receptor with Aβ (Mairet-Coello et al., 2013). Therefore, a neuronal receptor that is important in Aβ- induced tau pathology has not been identified and needs to be elucidated for controlling overall AD pathology. We added this information in ‘Introduction’ (fourth paragraph).

Then, how is FcγRIIb different from other Aβ receptors? It is reasonable to propose that different mechanisms or even the same mechanism exerts multiple effects at different stages of disease progression (De Strooper and Karran, 2016). For instance, RAGE is now believed to mainly function to transport Aβ in the blood brain barrier (Deane et al., 2003) and ABAD acts for mitochondrial toxicity as an intracellular binding partner of Aβ (Lustbader et al., 2004). In case of PrPc, it's debatable whether it is involved in Aβ-induced memory impairments and thus needs to be further characterized (Balducci et al., 2010; Gimbel et al., 2010; Cisse et al., 2011b). Further, compared to those receptors, our observations that the phosphorylation of FcγRIIb at tyrosine 273 is found in the brain of AD patients and is required for both oligomeric Aβ neurotoxicity and tau hyperphosphorylation can make it distinct from other Aβ- binding receptors. In the case of PirB that shares structural similarity with FcγRIIb and also acts as an Aβ receptor for synaptic plasticity, the phosphorylation of PirB is not associated with Aβ signaling (Kim et al., 2013). Thus, we believe that FcγRIIb facilitates tau phosphorylation and neuronal loss in AD brains, consistent with the proposed role of tau in AD pathogenesis, such as severe memory impairment and neuronal loss (Ballatore et al., 2007). Following the reviewer’s suggestion, we added the specificity of FcγRIIb in ‘Discussion’ (fourth paragraph).

In general, Aβ oligomers play a key role in AD pathogenesis, while soluble Aβ oligomers are still heterogeneous, including low or high *n* oligomers, and the proposals on which species of Aβ oligomers are responsible for the pathogenesis are a little in debate (reviewed in Benilova et al., 2012). We have here used 3 different sources of Aβ oligomers; synthetic Aβ oligomers, naturally secreted Aβ oligomers (7PA2 cells), and Aβ of 3xTg-AD model mice. Although synthetic Aβ oligomers are well-characterized and have been used widely for neurotoxicity, the acting concentration of synthetic Aβ (μM range) is relatively higher than that in AD brains. Compared to synthetic Aβ oligomers, conditioned medium from 7PA2 cells mainly contains not only Aβ dimers and trimers but also the different pools of Aβ oligomers, including zeta peptide, and show more potent neurotoxic properties (nM range) (Qi-Takahara et al., 2005; Haass and Selkoe, 2007). Moreover, the oligomers generated in 3xTg-AD mouse brain may be more complicated and needs to be identified. Nonetheless, we propose here that FcγRIIb plays a crucial role in tau phosphorylation and neurotoxicity in vitroand in vivoin response to these species of Aβ, probably a certain common species among the different sources of Aβ oligomers. We added above information in ‘Discussion’ (second paragraph).

*2) There are two major gaps in the proposed cascade and this should be mentioned at least in the Discussion to clarify this for the reader. First there are no mechanistic data in this work to link Aβ to FcγRIIb phosphorylation. The authors mention lyn kinase briefly (Discussion, third paragraph) but don't show any data. The second gap is a mechanistic link between increased PtdIns(3,4) and tau pathology. GSK-3-β kinase that catalyzes tau hyperphosphorylation, uses ATP as a donor of phosphate groups, so phosphoinositides are not a substrate. Therefore a possible link could be PtdIns(3,4)-mediated regulation of GSK-3β.*

The first gap; Lyn is a member of Src-family tyrosine kinases and catalyzes the phosphorylation of FcγRIIb at ITIM domain for the inhibition of leukocyte signaling (Smith and Clatworthy, 2010). Thus, we initially hypothesized that and tested whether FcγRIIb was phosphorylated by Lyn after treatment with Aβ. We found that Lyn was activated (phosphorylation at Tyr 397) by Aβ_1-42_ treatment in primary cortical neurons and SH-SY5Y neuronal cells (Figure 7). In addition, we observed that knockdown of Lyn expression by shRNA suppressed Aβ- induced Lyn activation and FcγRIIb phosphorylation (tyrosine 273) (Figure 7). To further address the role of Lyn in AD, we also isolated a novel inhibitor of Lyn from virtual library. When we examined the activity of this Lyn inhibitor in vitroand in vivo, we consistently found that the Lyn inhibitor effectively prevented FcγRIIb-induced cell death (Figure 7) and Aβ-induced Lyn activation and FcγRIIb phosphorylation (Figure 7). Moreover, Lyn inhibitor reduced the Aβ-induced cell death in primary cortical neurons (Figure 7). Taken together, Lyn is required for FcγRIIb phosphorylation at Tyr273 for Aβ neurotoxicity. We have actually prepared these results together with other data of in vivotests using AD models as a separate issue, and are thus sorry to say that we would like to show these results only to reviewers. It also needs more detail characterization. Please, understand us. Thank you very much.

Author response image 1.Lyn is required for FcγRIIb phosphorylation to mediate Ap neurotoxicity.(**A**) Primary cortical neurons (DIV 5) and SH-SY5Y cells were treated with PBS or 5 µM Aβ_1-42_ for 24 h. Cell lysates were prepared and subjected to western blotting. (**B**) SH-SY5Y cells were transfected with pSUPER-neo (Mock) or pLyn shRNA (shLyn) for 48 h and then incubated with PBS or 2 µM Aβ_1- 42_ for additional 12 h. Cell lysates were immunoblotted using the indicated antibodies. (**C**) HT22 cells were incubated with DMSO, 0.5 µM Lyn inhibitor or Scramble, and then transfected with pEGFP-N1 (Mock) or pFcγRIIb-EGFP (FcγRIIb) for 36 h. Cells showing fragmented nuclei after staining by EtHD were considered as dead cells. Values are means ± s.d. (n = 3). ***P<0.0005, **P<0.005, two-tailed *t*-test. (**D**) SH-SY5Y cells were pre-treated with the indicated concentrations (µM) of Lyn inhibitor or scramble for 2 h, and then transfected with pFcγRIIb-EGFP for additional 24 h. Cell lysates were immunoblotted. (**E**) Primary cortical neurons at DIV 5 were pre-treated with DMSO or the indicated concentrations of small molecules (Lyn inhibitor and scramble) for 2 h, and then incubated with 5 µM Aβ_1-42_ for 48 h. Values are means ± s.d. (n = 3). ^#^P<0.005 versus PBS with DMSO-treated sample. *P<0.05, **P<0.005 versus Aβ_1-42_ with DMSO-treated sample. two-tailed *t*-test.**DOI:**
http://dx.doi.org/10.7554/eLife.18691.022

The second gap; the previous study showed that GSK3β, a well-known tau kinase, is stimulated by ER stress (Ren et al., 2015). We previously demonstrated that ER stress is elicited by the interaction between Aβ_1-42_ and FcγRIIb, and is responsible for Aβ neurotoxicity (Kam et al., 2013). Thus, we decided to examine the gap among SHIP2, ER stress, and GSK3β. To determine the relationship between PtdIns(3,4)P_2_ (a SHIP2 product) and ER stress, we first analyzed the alteration of GRP78, a typical marker of unfolded protein response (UPR), following the delivery of various phosphoinositides into primary cortical neurons. Among the phosphoinositides, we found that PtdIns(3,4)P_2_ only increased the level of GRP78. We also found that PtdIns(3,4)P_2_ abolished the inhibitory phosphorylation of GSK3β at Ser9 (GSK3β activation) and increased tau hyperphosphorylation (Figure 5—figure supplement 2). In addition, we found that treatment with ER stress inhibitors, such as 4-PBA and Salubrinal, a chemical chaperone and an eIF2α dephosphorylation inhibitor, respectively, apparently attenuated PtdIns(3,4)P_2_-induced GSK3β activation and tau phosphorylation with some different efficacy (Figure 5—figure supplement 2). The different efficacy of these two inhibitors on GSK3β and tau phosphorylation might reflect a way of their inhibition of ER stress and tau phosphorylation by kinases. These results suggest that GSK3β is activated by SHIP2 product through ER stress to phosphorylate tau. Further, treatment with AS1949490, a SHIP2 inhibitor, or infection with lentivirus carrying *Ship2* shRNA blocked Aβ-induced GSK3β activation and tau hyperphosphorylation (Figure 5—figure supplement 2), again confirming our proposal that activation of SHIP2 elicits ER stress to activate GSK3β and subsequent tau hyperphosphorylation. We included these results into supplementary information (Figure 5—figure supplement 2) (subsection “Dysregulation of phosphoinositide metabolism by the FcγRIIb-SHIP2 axis for tau 269 phosphorylation”, last paragraph).

*3) Two referees have concerns with the idea that the action of FcγRIIb is cell autonomous. First there are issues with the expression of FcγRIIb in neurons: Figure 1—figure supplement 1 A shows in fact that a large part of FcγRIIb mRNA is in glia and not in neurons. Additional evidence to proof that FcγRIIb is expressed in neurons is needed, it could be that the weak signal in the experiments in neurons is background. The authors could consider in situ hybridisation to check their claim. In addition, FcγRIIb is highly expressed in microglial cells and may play a role in microglial activation. Therefore, the impact of FcγRIIb KO on tau pathology in 3X TG mice can be also mediated by reduced microglial activation (cell autonomous effects vs non-autonomous effects). Did the authors check microglial activation and cytokine levels in these mice (WT 3XTG vs. KO 3Xtg)? Moreover, have the authors looked at whether FcγRIIb plays a role in sensing Aβ in microglial cells?*

To evaluate neuronal expression of FcγRIIb, we have really paid great attention and by performing many experiments.

First, as already shown in our previous manuscript (Kam et al., 2013), FcγRIIb mRNA (RT-PCR) and proteins (western blot analysis and immunostaining) were found in the neurons of mouse and AD patient brains (Figure 8). Moreover, unlike other FcγRs, levels of FcγRIIb mRNA (RT-PCR) and proteins were greatly increased at least 3 to 10 folds by Aβ treatment in primary cultured neurons and NeuN-positive neurons in AD brains. For these analyses, we tested almost all of commercially available and home-made FcγRIIb antibodies (more than 7 antibodies) and consistently found similar results using a few of the antibodies.

Author response image 2.Expression of FcγRIIb protein and mRNA in the purified neurons.(**C**, **D**) Detection of FcγRIIb protein in NeuN- positive neurons of normal and AD patients by immunohistochemistry. (**E**) Detection of FcγRIIb protein by western blotting in the primary cultured cortical neurons. (**F**) Increase in *Fcgr2b* mRNA by Aβ_1-42_ in primary cortical neurons. RT-PCR analysis was performed with synthetic primers.**DOI:**
http://dx.doi.org/10.7554/eLife.18691.023

Second, interestingly, we found a neuron-enriched isoform of FcγRIIb, FcγRIIb2, in the purified neurons and whole lysate of mouse hippocampus and cortex, whereas spleen and thymus express other isoform of FcγRIIb, FcγRIIb1 (Figure 9). Also, it is *Fcgr2b2* mRNA that is selectively increased by oligomeric Aβ_1-42_ treatment in primary cortical neurons (Figure 9). From DNA sequencing analysis, we found that FcγRIIb2 is lacking one exon (IE1) in its cytoplasmic domain and exhibits a distinct function from FcγRIIb1, especially in neurotoxic activity. We are submitting a manuscript showing their functional differences in the neuropathologic activity in AD model to other journal and decided to show the results only to reviewer to support our current proposal.

Author response image 3.Detection of neuron-enriched FcetIIb isoform, FcγRIIb2.(**A**) RT-PCR analysis showing *Fcgr2b* mRNA in the purified neurons of WT and Fcgr2b KO brain and other tissues. (**B**) Increase of *Fcgr2b2* mRNA by oligomeric Aβ_1-42_, but not by monomeric Aβ_1-42_, in primary cortical neurons.**DOI:**
http://dx.doi.org/10.7554/eLife.18691.024

Third, to evaluate neuronal activity of FcγRIIb, we know that generation of neuron- specific conditional knockout (nsKO) mice of FcγRIIb is the best. However, as you known well, generation of neuron-specific knockout (nsKO) mice of FcγRIIb and of FcγRIIb nsKO/3x AD double transgenic mice will take too long time (3 more years). Thus, as an alternative approach, we generated a dominant negative mutant of FcγRIIb (FcγRIIb DN; FcγRIIb-ΔCyto) lacking its cytoplasmic region (a.a. 232-300) and thus unable to transmit the signal into cells, and confirmed its protective activity in Aβ-induced neuronal cell death after its transient expression in neuronal cells (data not shown). We then generated a transgenic mouse expressing FcγRIIb- ΔCyto-HA under *CamKIIα* promoter, which drives neuron-specific expression of the transgene (Mayford et al., 1996) (Figure 10). We confirmed apparent but low expression of FcγRIIb-ΔCyto-HA in the hippocampus and cortex of the transgenic mice (Figure 10). To address neuropathic effect FcγRIIb in AD model mice, we next crossed the mice with 3xTg-AD model mice to generate 3xTg- AD/FcγRIIb-ΔCyto-HA double transgenic mice (Figure 10). The behavior tests of the mice showed that neuronal expression of FcγRIIb-ΔCyto-HA inhibited memory impairments of 3xTg-AD model mice in Y-maze, novel object recognition, and passive avoidance tests (Figure 10). These results are essentially similar with the results observed in 3xTg-AD/FcγRIIb KO double transgenic mice, and suggest that neuronal function of FcγRIIb is critical for the memory impairments in the AD model mice. As we mentioned above, these results together with those of neuron-enriched FcγRIIb2 isoform are included in other manuscript and are thus presented only here to provide more convincing evidence to reviewer.

Author response image 4.Neuron-specific effect of FcγRIIb-dominant negative mutant on memory function in the transgenic mice.(**A**) Construction of CamKIIα promoter-FcγRIIb deletion mutant lacking its cytoplasmic domain (∆232-300) (FcγRIIb DN). Arrows indicate PCR primers for genotyping of the mice. (**B**) Western blotting showing neuronal expression of FcγRIIb-DN in the transgenic mice. n.s. non-specific signals. (**C**) Genomic PCR analysis showing generation of 3xTg-AD/FcγRIIb DN double transgenic mice. (**D**) Behavior tests showing memory rescue in the 3xTg-AD mice by FcγRIIb DN expression. (left) Y-maze test, (middle) noble object recognition test, (right) passive avoidance test. (n = 9-11 male mice per group). Values are means ± s.e.m.; *P<0.05, **P<0.01, ***P<0.005, unpaired *t*-test.**DOI:**
http://dx.doi.org/10.7554/eLife.18691.025

Fourth, following the reviewer’s suggestion, we searched and found the neuronal expression of murine FcγRIIb by the in situhybridization (ISH) results provided from Allen Brain Atlas (http://www.brain-map.org/). Although *fcgR2b* expression is quite low, *fcgr2b*-positive ISH signals were detected in the hippocampus of P56 male mouse (Figure 11).

Author response image 5.Expression pattern of *Fcgr2b* in mouse hippocampus from Allen Brain Atlas.Website: © 2015 Allen Institute for Brain Science. Allen Brain Atlas [Internet]. Available from: http://www.brain-map.org. In situ hybridization (**A**) and expression image (**B**) of *Fcgr2b* in sagittal section of brain at P56 were displayed. Probes that span from exon 1 to exon 7 of *Fcgr2b* transcript were used. Arrows indicate the positive signals located in either pyramidal cells in CA1 and CA3 or granule cells in dentate gyrus (DG). Bar, 200 µm.**DOI:**
http://dx.doi.org/10.7554/eLife.18691.026

Finally, consistent to our proposal, there are references showing the expression of FcγRIIb mRNA and proteins in neurons:

(1) Cahoy et al. [J. Neurosci. (2008)] created a transcriptome database of the expression levels of >20,000 genes by gene profiling from astrocytes, neurons and oligodendrocytes. Based on their list, FcγRIIb is significantly expressed in postnatal day 16 (P16) neurons and P17 forebrain. (2) Suemitsu et al. [Neuroscience (2010)] detected FcγRIIb protein in parvalbumin neurons, whereas FcγRIII and FcγRI proteins were detected in microglial cells.

In summary, we confirmed that, though neuronal expression is very low compared to non-neuronal cells, FcγRIIb is evidently expressed in neuronal cells as well. Again, please note that its expression is increased at least several folds in primary neurons by Aβ treatment and in AD brains compared to age-matched non- AD brains. Moreover, presence and distinct function of a neuron-enriched FcγRIIb2 provides more compelling evidence for the pathogenic role of FcγRIIb in AD.

While we have mainly focused on the role of neuronal FcγRIIb in tau pathology of AD model, we were also attempted to examine the role of FcγRIIb in the microglial activation in 3x Tg-AD mice. From immunohistochemical analysis, we identified that, compared to that of age-matched control mice, expression of Iba1, a microglia marker, was significantly reduced by *Fcgr2b* KO in the cortex (Figure 12) and slightly in the hippocampus (Figure 12) of 3x Tg-AD mice. In addition, RT-PCR analysis showed that mRNAs of pro-inflammatory genes (IL1β, iNOS, and TNFα) were also decreased by *Fcgr2b* KO in 3x Tg-AD mice (Figure 12). Though there is a tissue-specific difference, these observations suggest that ablation of FcγRIIb might reduce microglia activation in the brain of 3x Tg-AD mice. Unlike neuron-specific effects of FcγRIIb-ΔCyto-HA on tau phosphorylation and memory impairment seen in a transgenic mice expressing FcγRIIb-ΔCyto-HA, however, we do not think that the reduced inflammation by *Fcgr2b* KO in 3x Tg-AD mice is a direct cause of the memory rescue. At this moment, we do believe that the neuroinflammation via FcγRIIb also functions to contribute to the aggravation of memory impairment in AD mice, while we do not have evidence showing that the non-neuronal activity of FcγRIIb is solely responsible for the memory impairment. This role of FcγRIIb in the regulation of neuroinflammation is important and is worthwhile to pursue further in detail but is far away from our main story at this time. We showed these only to reviewers and discussed it in ‘Discussion’ as ‘data not shown’ (third paragraph).

Author response image 6.Microglial activation is suppressed by *Fcgr2b* KO in 3x Tg-AD mice.(**A**) Immunohistochemical analysis of Iba1 levels in the cortex and hippocampus of 7 month-old 3x Tg-AD and 3x Tg-AD/*Fcgr2b* KO mice. Magnifying power 200x. White scale bar, 20 µm. (B, C) Quantification of Iba1 immunoreactivity in the cortex (**B**) and hippocampus (**C**) of 3x Tg-AD and 3x Tg-AD/*Fcgr2b* KO mice. Values are means ± s.e.m. (n = 8). *P<0.05 compared with 3x Tg-AD mice. (**D**) Expression of IL1β, iNOS, and TNFα were diminished in 3x Tg-AD/*Fcgr2b* KO mice. Total RNAs were extracted from cortical lysates from 6 month-old 3x Tg-AD and 3x Tg-AD/*Fcgr2b* KO mice and were analyzed by quantitative RT-PCR. Values are means ± s.e.m. (n = 3). **P*<0.05.**DOI:**
http://dx.doi.org/10.7554/eLife.18691.027

*4) The authors should provide in the next version the pictures of Figure 1—figure supplement 2 which are now missing. These data are quite important as demonstrating the effect of the FcγRIIb receptor deficiency on APP processing.*

We are sorry for missing the pictures. We added the figures and separated those into Figure 1—figure supplement 2 and Figure 1—figure supplement 3 in the revised manuscript)

*5) In the last paragraph of the subsection “FcJRIIb is essential for tau hyperphosphorylation and memory deficit in 3xTg-AD mice”, it is said that 9months old 3xTgAD mouse are "plaque-free". What happens in the cortex of these 3xTg mice where Aβ plaques should be present, is tau phosphorylation more pronounced there? And is that protected by FcγRIIb deficiency? Does the age of mice affect tau hyperphoshorylation, i.e. is tau pathology still absent in 3xTg FcγRIIb KO mouse even at a very advanced age when there are loads of Aβ?*

In general, it is known that extracellular amyloid plaques were prominently detected over 12 month of age in 3x Tg-AD mouse model (Hirata-Fukae et al., 2008). Since we mostly analyzed tau phosphorylation at the age of 8-9 months, we performed additional experiments to identify the effect of FcγRIIb KO in more aged mice following the reviewer’s suggestion. While transgenic expression of human tau (detected by HT7 antibody) in 3x Tg-AD mice was not altered during aging, tau phosphorylation in 20 month-old 3x Tg-AD mice was heavily detected and occurred more than that of 6 month-old 3x Tg-AD mice (Figure 1—figure supplement 3). Compared to 15 month-old 3x Tg-AD mice, the level of hyperphosphorylated tau was reduced by *Fcgr2b* KO in the same age of 3x Tg-AD mice (Figure 1—figure supplement 3). These results illustrate that tau phosphorylation mediated by FcγRIIb occurs at the onset of the disease and lasts to the late stage. We included these results (Figure 1—figure supplement 3) (subsection “FcγRIIb is essential for tau hyperphosphorylation and memory deficit in 3xTg-AD mice”, last paragraph).

*6) In Figure 3—figure supplement 2 the authors analyze the effects of the receptor mutations on Aβ-induced cell death. "Cell death was determined after 24h under fluorescence microscope". What method/dye/criterion was used? How many spots were analyzed per sample per condition to produce statistics on panel B?*

Apoptotic cell death was determined by morphology-based death assay as described in the previous literatures (Song et al., 2008; Han et al., 2010). After transfection with the DNAs, GFP-positive cells showing condensed and fragmented nuclei after staining with Hoechst 33258 were counted under a fluorescence microscope. We counted at least 900 cells in 3~4 random spots per sample per condition. We added this information into the figure legend of the revised manuscript (Figure 3—figure supplement 2 legend). To confirm this result of cell viability, we additionally determined their neurotoxicity using other cell viability assay in HT22 cells. We observed similar results as those mentioned above (Figure 13).

Author response image 7.Association of the neurotoxic activity of FcγRIIb with ITIM.HT22 cells were transiently transfected with pEGFP (Mock), FcγRIIb-GFP (WT), FcγRIIb ITIM-deleted form (1-270; ΔITIM), FcγRIIb cytosolic domain-deleted form (1-240; ΔCyto), or FcγRIIb phosphorylation-defective mutant in ITIM (Y273F) for 24 h. Cell viability were determined by using EZ-CYTOXTM cell viability kit Values are means ± s.e.m. (n = 3). ***P*<0.01, N.S., not significant, compared with Mock control.**DOI:**
http://dx.doi.org/10.7554/eLife.18691.028